# WHY FLATNESS DOES AND DOES NOT CORRELATE WITH GENERALIZATION FOR DEEP NEURAL NETWORKS

## ABSTRACT

The intuition that local flatness of the loss landscape is correlated with better generalization for deep neural networks (DNNs) has been explored for decades, spawning many different flatness measures. Recently, this link with generalization has been called into question by a demonstration that many measures of flatness are vulnerable to parameter re-scaling which arbitrarily changes their value without changing neural network outputs. Here we show that, in addition, some popular variants of SGD such as Adam and Entropy-SGD, can also break the flatness-generalization correlation. As an alternative to flatness measures, we use a function based picture and propose using the log of Bayesian prior upon initialization, $\log P(f)$, as a predictor of the generalization when a DNN converges on function $f$ after training to zero error. The prior is directly proportional to the Bayesian posterior for functions that give zero error on a training set. For the case of image classification, we show that $\log P(f)$ is a significantly more robust predictor of generalization than flatness measures are. Whilst local flatness measures fail under parameter re-scaling, the prior/posterior, which is global quantity, remains invariant under re-scaling. Moreover, the correlation with generalization as a function of data complexity remains good for different variants of SGD.

## 1 INTRODUCTION

Among the most important theoretical questions in the field of deep learning are: 1) What characterizes functions that exhibit good generalization?, and 2) Why do overparameterized deep neural networks (DNNs) converge to this small subset of functions that do not overfit? Perhaps the most popular hypothesis is that good generalization performance is linked to flat minima. In pioneering works (Hinton & van Camp, 1993; Hochreiter & Schmidhuber, 1997), the minimum description length (MDL) principle (Rissanen, 1978) was invoked to argue that since flatter minima require less information to describe, they should generalize better than sharp minima. Most measures of flatness approximate the local curvature of the loss surface, typically defining flatter minima to be those with smaller values of the Hessian eigenvalues (Keskar et al., 2016; Wu et al., 2017; Zhang et al., 2018; Sagun et al., 2016; Yao et al., 2018).

Another commonly held belief is that stochastic gradient descent (SGD) is itself biased towards flatter minima, and that this inductive bias helps explain why DNNs generalize so well (Keskar et al., 2016; Jastrzebski et al., 2018; Wu et al., 2017; Zhang et al., 2018; Yao et al., 2018; Wei & Schwab, 2019; Maddox et al., 2020). For example Keskar et al. (2016) developed a measure of flatness that they found correlated with improved generalization performance when decreasing batch size, suggesting that SGD is itself biased towards flatter minima. We note that others (Goyal et al., 2017; Hoffer et al., 2017; Smith et al., 2017; Mingard et al., 2021a) have argued that the effect of batch size can be compensated by changes in learning rate, complicating some conclusions from Keskar et al. (2016). Nevertheless, the argument that SGD is somehow itself biased towards flat minima remains widespread in the literature.

In an important critique of local flatness measures, Dinh et al. (2017) pointed out that DNNs with ReLU activation can be re-parameterized through a simple parameter-rescaling transformation.

$$T_\alpha : (\mathbf{w}_1, \mathbf{w}_2) \mapsto \left(\alpha \mathbf{w}_1, \alpha^{-1} \mathbf{w}_2\right) \tag{1}$$

where $\mathbf{w}_1$ are the weights between an input layer and a single hidden layer, and $\mathbf{w}_2$ are the weights between this hidden layer and the outputs. This transformation can be extended to any architecture having at least one single rectified network layer. The function that the DNN represents, and thus how it generalizes, is invariant under parameter-rescaling transformations, but the derivatives w.r.t. parameters, and therefore many flatness measures used in the literature, can be changed arbitrarily. *Ergo*, the correlation between flatness and generalization can be arbitrarily changed.

Several recent studies have attempted to find "scale invariant" flatness metrics (Petzka et al., 2019; Rangamani et al., 2019; Tsuzuku et al., 2019). The main idea is to multiply layer-wise Hessian eigenvalues by a factor of $\|\mathbf{w_i}\|^2$, which renders the metric immune to layer-wise re-parameterization. While these new metrics look promising experimentally, they are only scale-invariant when the scaling is layer-wise. Other methods of rescaling (e.g. neuron-wise rescaling) can still change the metrics, so this general problem remains unsolved.

## 1.1 MAIN CONTRIBUTIONS

1. For a series of classic image classification tasks (MNIST and CIFAR-10) we show that flatness measures change substantially as a function of epochs. Parameter re-scaling can arbitrarily change flatness, but it quickly recovers to a more typical value under further training. We also demonstrate that some variants of SGD exhibit significantly worse correlation of flatness with generalization than found for vanilla SGD. In other words popular measures of flatness sometimes do and sometimes do not correlate with generalization. This mixed performance problematizes a widely held intuition that DNNs generalize well fundamentally because SGD or its variants are themselves biased towards flat minima.

2. We next study the correlation of the Bayesian prior $P(f)$ with the generalization performance of a DNN that converges to that function $f$. This prior is the weighted probability of obtaining function $f$ upon random sampling of parameters. Motivated by a theoretical argument derived from a non-uniform convergence generalization bound, we show empirically that $\log P(f)$ correlates robustly with test error, even when local flatness measures miserably fail, for example upon parameter re-scaling. For discrete input/output problems (such as classification), $P(f)$ can also be interpreted as the weighted "volume" of parameters that map to $f$. Intuitively, we expect local flatness measures to typically be smaller (flatter) for systems with larger volumes. Nevertheless, there may also be regions of parameter space where local derivatives and flatness measures vary substantially, even if on average they correlate with the volume. Thus flatness measures can be viewed as (imperfect) local measures of a more robust predictor of generalization, the volume/prior $P(f)$.

## 2 DEFINITIONS AND NOTATION

### 2.1 SUPERVISED LEARNING

For a typical supervised learning problem, the *inputs* live in an input domain $\mathcal{X}$, and the *outputs* belong to an output space $\mathcal{Y}$. For a *data distribution* $\mathcal{D}$ on the set of input-output pairs $\mathcal{X} \times \mathcal{Y}$, the *training set* $S$ is a sample of $m$ input-output pairs sampled i.i.d. from $\mathcal{D}$, $S = \{(x_i, y_i)\}_{i=1}^{m} \sim \mathcal{D}^m$, where $x_i \in \mathcal{X}$ and $y_i \in \mathcal{Y}$. The output of a DNN on an input $x_i$ is denoted as $\hat{y}_i$. Typically a DNN is parameterized by a vector $\mathbf{w}$ and trained by minimizing a *loss function* $L : \mathcal{Y} \times \mathcal{Y} \to \mathbb{R}$, which measures differences between the output $\hat{y} \in \mathcal{Y}$ and the ground truth output $y \in \mathcal{Y}$, by assigning a score $L(\hat{y}, y)$ which is typically zero when they match, and positive when they don't match. Minimizing the loss typically involves using an optimization algorithm such as SGD on a training set $S$. The generalization performance of the DNN, which is theoretically defined over the underlying (typically unknown) data distribution $\mathcal{D}$ but is practically measured on a *test set* $E = \{(x'_i, y'_i)\}_{i=1}^{|E|} \sim \mathcal{D}^{|E|}$. For classification problems, the *generalization error* is practically measured as $\epsilon(E) = \frac{1}{|E|} \sum_{x'_i \in E} \mathbb{1}[\hat{y}_i \neq y'_i]$, where $\mathbb{1}$ is the standard indicator function which is one when its input is true, and zero otherwise.

## 2.2 Flatness measures

Perhaps the most natural way to measure the flatness of minima is to consider the eigenvalue distribution of the Hessian $H_{ij} = \partial^2 L(\mathbf{w})/\partial w_i \partial w_j$ once the learning process has converged (typically to a zero training error solution). Here for simplicity we use $L(\mathbf{w})$ instead of $L(\hat{y}, y)$ as $\hat{y}$ is parameterized by $\mathbf{w}$. Sharp minima are characterized by a significant number of large positive eigenvalues $\lambda_i$ in the Hessian, while flat minima are dominated by small eigenvalues. Some care must be used in this interpretation because it is widely thought that DNNs converge to stationary points that are not true minima, leading to negative eigenvalues and complicating their use in measures of flatness. Typically, only a subset of the positive eigenvalues are used (Wu et al., 2017; Zhang et al., 2018). Hessians are typically very expensive to calculate. For this reason, Keskar et al. (2016) introduced a computationally more tractable measure called "sharpness":

**Definition 2.1** (Sharpness). Given parameters $\mathbf{w}'$ within a box in parameter space $\mathcal{C}_\zeta$ with sides of length $\zeta > 0$, centered around a minimum of interest at parameters $\mathbf{w}$, the sharpness of the loss $L(\mathbf{w})$ at $\mathbf{w}$ is defined as:

$$\text{sharpness} := \frac{\max_{\mathbf{w}' \in \mathcal{C}_\zeta} (L(\mathbf{w}') - L(\mathbf{w}))}{1 + L(\mathbf{w})} \times 100.$$

In the limit of small $\zeta$, the sharpness relates to the spectral norm of the Hessian (Dinh et al., 2017):

$$\text{sharpness} \approx \frac{\left\| \left| \left( \nabla^2 L(\mathbf{w}) \right) \right| \right\|_2 \zeta^2}{2(1 + L(\mathbf{w}))} \times 100.$$

The general concept of flatness can be defined as $1/sharpness$, and that is how we will interpret this measure in the rest of this paper.

## 2.3 Functions and the Bayesian prior

We first clarify how we represent functions in the rest of paper using the notion of *restriction of functions*. A more detailed explanation can be found in appendix C. Here we use binary classification as an example:

**Definition 2.2** (Restriction of functions to $C$). (Shalev-Shwartz & Ben-David, 2014) Consider a parameterized supervised model, and let the input space be $\mathcal{X}$ and the output space be $\mathcal{Y}$, noting $\mathcal{Y} = \{0, 1\}$ in binary classification setting. The space of functions the model can express is a (potentially uncountably infinite) set $\mathcal{F} \subseteq \mathcal{Y}^{|\mathcal{X}|}$. Let $C = \{c_1, \ldots, c_m\} \subset \mathcal{X}$. The restriction of $\mathcal{F}$ to $C$ is the set of functions from $C$ to $\mathcal{Y}$ that can be derived from functions in $\mathcal{F}$:

$$\mathcal{F}_C = \{(f(c_1), \ldots, f(c_m)) : f \in \mathcal{F}\}$$

where we represent each function from $C$ to $\mathcal{Y}$ as a vector in $\mathcal{Y}^{|C|}$.

For example, for binary classification, if we restrict the functions to $S + E$, then each function in $\mathcal{F}_{S+E}$ is represented as a binary string of length $|S| + |E|$. In the rest of paper, we simply refer to "functions" when we actually mean the restriction of functions to $S + E$, except for the Boolean system in section 5.1 where no restriction is needed. See appendix C for a thorough explanation.

For discrete functions, we next define the prior probability $P(f)$ as

**Definition 2.3** (Prior of a function). Given a prior parameter distribution $P_w(\mathbf{w})$ over the parameters, the *prior of function $f$* can be defined as:

$$P(f) := \int \mathbb{1}[\mathcal{M}(\mathbf{w}) = f] P_w(\mathbf{w}) d\mathbf{w}. \tag{2}$$

where $\mathbb{1}$ is an indicator function: $\mathbb{1}[arg] = 1$ if its argument is true or 0 otherwise; $\mathcal{M}$ is the parameter-function map whose formal definition is in appendix B. Note that $P(f)$ could also be interpreted as a weighted volume $V(f)$ over parameter space. If $P_w(\mathbf{w})$ is the distribution at initialization, the $P(f)$ is the prior probability of obtaining the function at initialization. We normally use this parameter distribution when interpreting $P(f)$.

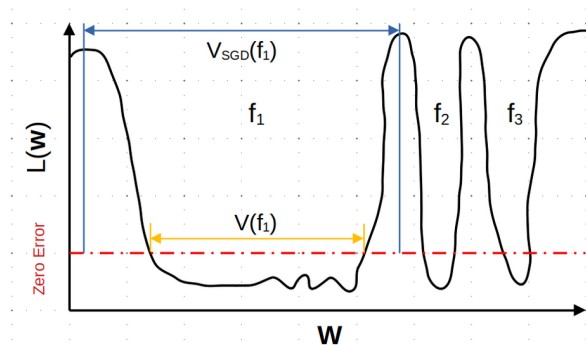

Figure 1: **Schematic loss landscape for three functions that have zero-error on the training set.** It illustrates how the relative sizes of the volumes of their basins of attraction $V_{\mathrm{SGD}}(f_i)$ correlate with the volumes $V(f_i)$ (or equivalently their priors $P(f_i)$) of the basins, and that, on average, larger $V(f_i)$ or $P(f_i)$ implies flatter functions, even if flatness can vary locally. Note that the loss $L(\mathbf{w})$ can vary within a region where the DNN achieves zero classification error on $S$. A similar schematic plot can be seen in (Mingard et al., 2021b), but here it is more clear that local flatness can be misleading. Note the "error" means the classification error, which is different from the loss.

*Remark.* Definition 2.3 works in the situation where the space $\mathcal{X}$ and $\mathcal{Y}$ are discrete, where $P(f)$ has a prior probability mass interpretation. This is enough for most image classification tasks. Nevertheless, we can easily extend this definition to the continuous setting, where we can also define a *prior density* over functions upon random initialization, with the help of Gaussian Process (GP) (Rasmussen, 2003). For the GP prior see appendix D. However, in this work, we focus exclusively on the classification setting, with discrete inputs and outputs.

## 2.4 LINK BETWEEN THE PRIOR AND THE BAYESIAN POSTERIOR

Due to their high expressivity, DNNs are typically trained to zero training error on the training set $S$. In this case the Bayesian picture simplifies because if functions are conditioned on zero error on $S$, this leads to a simple 0-1 *likelihood* $P(S|f)$, indicating whether the data is consistent with the function. Bayesian inference can be used to calculate a Bayesian *posterior probability* $P_B(f|S)$ for each $f$ by conditioning on the data according to Bayes rule. Formally, if $S = \{(x_i, y_i)\}_{i=1}^m$ corresponds to the set of training pairs, then

$$P_B(f|S) = \begin{cases} P(f)/P(S) \text{ if } \forall i, \ f(x_i) = y_i \\ 0 \text{ otherwise .} \end{cases}$$

where $P(f)$ is the Bayesian prior and $P(S)$ is called the *marginal likelihood* or *Bayesian evidence*. If we define, the training set neutral space $\mathcal{N}_S$ as all parameters that lead to functions that give zero training error on $S$, then $P(S) = \int_{\mathcal{N}_S} P_{\mathbf{w}}(\mathbf{w}) d\mathbf{w}$. In other words, it is the total prior probability of all functions compatible with the training set $S$ (Valle-Pérez et al., 2018; Mingard et al., 2021a). Since $P(S)$ is constant for a given $S$, $P_B(f|S) \propto P(f)$ for all $f$ consistent with that $S$.

## 3 THE CORRELATION BETWEEN THE PRIOR AND GENERALIZATION

This link between the prior and the posterior is important, because it was empirically found in an extensive set of experiments by (Mingard et al., 2021a) that, for popular architectures and data sets,

$$P_B(f|S) \approx P_{\mathrm{SGD}}(f|S), \tag{3}$$

where $P_{\mathrm{SGD}}(f|S)$ is the probability that a DNN trained with SGD converges on function $f$, when trained to zero error on $S$. In other words, to first order, SGD appears to find functions with a probability predicted by the Bayesian posterior, and thus with probabilities directly proportional to $P(f)$. The authors traced this behaviour to the geometry of the loss landscape, as follows. Some general observations from algorithmic information theory (AIT) (Valle-Pérez et al., 2018) as well

as direct calculations (Mingard et al., 2019) predict that the priors of functions should vary over many orders of magnitude. When this is the case, it is reasonable to expect that the probabilities by which an optimizer finds different functions is affected by these large differences. This is related to a mechanism identified previously in evolutionary dynamics, where it is called the arrival of the frequent (Schaper & Louis, 2014). We illustrate this principle in fig. 1 where we intuitively use the language of "volumes". We expect that the relative sizes of the basins of attraction $V_{SGD}(f)$, defined as the set of initial parameters for which a DNN converges to a certain function $f$, is proportional, to first order, to those of the priors $P(f)$ (or equivalently the "volumes"). To second order there are, of course, many other features of a search method and a landscape that affect what functions a DNN converges on, but when the volumes/priors vary by so many orders of magnitude then we expect that to first order $P_{SGD}(f|S) \approx P_B(f|S) \propto P(f) = V(f)$.

Given that the $P(f)$ of a function helps predict how likely SGD is to converge on that function, we can next ask how $P(f)$ correlates with generalization. Perhaps the simplest argument is that if DNNs trained to zero error are known to generalize well on unseen data, then the probability of converging on functions that generalize well must be high. The $P(f)$ of these functions must be larger than the priors of functions that do not generalize well. Can we do better than this rather simplistic argument? One way forward is empirical. Mingard et al. (2021a) showed that $\log(P_B(f|S))$ correlates quite tightly with generalization error. These authors also made a theoretical argument based on the Poisson-Binomial nature of the error distribution to explain this log-linear relationship, but this approach needs further work.

One of the best overall performing predictors in the literature for generalization performance on classification tasks is the marginal likelihood PAC-Bayes bound from Valle-Pérez et al. (2018); Valle-Pérez & Louis (2020). It is non-vacuous, relatively tight, and can capture important trends in generalization performance with training set size (learning curves), data complexity, and architecture choice (see also (Liu et al., 2021)). However, the prediction uses the marginal likelihood $P(S)$ defined through a sum over all functions that produce zero error on the training set. Here we are interested in the generalization properties of single functions. One way around is to use a simple nonuniform bound which to the best of our knowledge was first published in McAllester (1998) as a preliminary theory to the full PAC-Bayes theorems. For any countable function space $\mathcal{F}$, any distribution $\tilde{P}$, and for any selection of a training set $S$ of size $m$ under probability distribution $\mathcal{D}$, it can be proven that for all functions $f$ that give zero training error:

$$\forall \mathcal{D}, \mathbf{P}_{S \sim \mathcal{D}^m} \left[ \epsilon_{S,E}(f) \leq \frac{\ln \frac{1}{\tilde{P}(f)} + \ln \frac{1}{\delta}}{m} \right] \geq 1 - \delta \tag{4}$$

for $\delta \in (0,1)$. Here we consider a space $\mathcal{F}_{S,E}$ of functions with all possible outputs on the inputs of a specific $E$ and zero error on a specific $S$; $\epsilon_{S,E}(f)$ is the error measured on $E + S$, which as the error on $S$ is 0, equals the error on the test set $E$. This error will converge to the true generalization error on all possible inputs as $|E|$ increases. Valle-Pérez & Louis (2020) showed this bound has an optimal average generalization error when $\tilde{P}(f)$ mimics the probability distribution over functions of the learning algorithm. If $P_{SGD}(f) \approx P_B(f|S) \propto P(f)$, then the best performance of the bound is approximately when $\tilde{P}(f)$ in eq. (4) is the Bayesian prior $P(f)$. Thus this upper bound on $\epsilon_{S,E}(f)$ scales as $-\log(P(f))$.

## 4 FLATNESS, PRIORS AND GENERALIZATION

The intuition that larger $P(f)$ correlates with greater flatness is common in the literature, see e.g. Hochreiter & Schmidhuber (1997); Wu et al. (2017), where the intuition is also expressed in terms of volumes. If volume/$P(f)$ correlates with generalization, we expect flatness should too. Nevertheless, local flatness may still vary significantly across a volume. For example Izmailov et al. (2018) show explicitly that even in the same basin of attraction, there can be flatter and sharper regions. We illustrate this point schematically in fig. 1, where one function clearly has a larger volume and on average smaller derivatives of the loss w.r.t. the parameters than the others, and so is flatter on average. But, there are also local areas within the zero-error region where this correlation does not hold. One of the main hypotheses we will test in this paper is that the correlation between flatness and generalization can be broken even when the generalization-prior correlation remains robust.

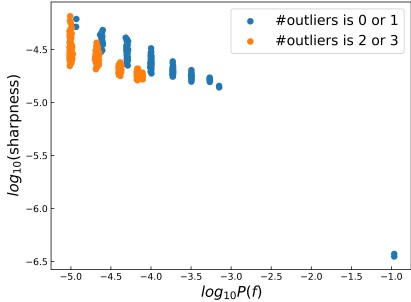

Figure 2: **The correlation between flatness and the Bayesian prior for the $n = 7$ Boolean system.** The functions are defined on the full space of 128 possible inputs. The priors $P(f)$ are shown for the 1000 most frequently found functions by SGD from random initialization for a two hidden layer FCN, and correlate well with $\log$(flatness). The function with the largest prior, which is the most "flat" is the trivial one of all 0s or all 1s. An additional feature is two offset bands caused by a discontinuity of Boolean functions. Most functions shown are mainly 0s or mainly 1s, and the two bands correspond to different number of outliers (e.g. 1s when the majority is 0s).

## 5 EXPERIMENTAL RESULTS

### 5.1 PRIOR/VOLUME - FLATNESS CORRELATION FOR BOOLEAN SYSTEM

We first study a model system for Boolean functions of size $n = 7$, which is small enough to directly measure the prior by sampling (Valle-Pérez et al., 2018). There are $2^7 = 128$ possible binary inputs. Since each input is mapped to a single binary output, there are $2^{128} = 3.4 \times 10^{34}$ possible functions $f$. It is only practically possible to sample the prior $P(f)$ because it is highly biased (Valle-Pérez et al., 2018; Mingard et al., 2019), meaning a subset of functions have priors much higher than average. For a fully connected network (FCN) with two hidden layers of 40 ReLU units each (which was found to be sufficiently expressive to represent almost all possible functions) we empirically determined $P(f)$ using $10^8$ random samples of the weights $\mathbf{w}$ over an initial Gaussian parameter distribution $P_w(\mathbf{w})$ with standard deviation $\sigma_w = 1.0$ and $\sigma_b = 0.1$.

We also trained our network with SGD using the same initialization and recorded the top-1000 most commonly appearing output functions with zero training error on all 128 outputs, and then evaluated the sharpness/flatness using definition 2.1 with an $\epsilon = 10^{-4}$. For the maximization process in calculating sharpness/flatness, we ran SGD for 10 epochs and make sure the max value ceases to change. As fig. 2 demonstrates, the flatness and prior correlate relatively well; fig. S8 in the appendix shows a very similar correlation for the spectral norm of the Hessian. Note that since we are studying the function on the complete input space, it is not meaningful to speak of correlation with generalization. However, since for this system the prior $P(f)$ is known to correlate with generalization (Mingard et al., 2021a), the correlation in fig. 2 also implies that these flatness measures will correlate with generalization, at least for these high $P(f)$ functions.

### 5.2 PRIORS, FLATNESS AND GENERALIZATION FOR MNIST AND CIFAR-10

We next study the correlation between generalization, flatness and $\log P(f)$ on the real world datasets MNIST (LeCun et al., 1998) and CIFAR-10 (Krizhevsky et al., 2009). Because we need to run many different experiments, and measurements of the prior and flatness are computationally expensive, we simplify the problem by binarizing MNIST (one class is 0-4, the other is 5-9) and CIFAR-10 (we only study two categories out of ten: cars and cats). Also, our training sets are relatively small (500/5000 for MNIST/CIFAR-10, respectively) but we have checked that our overall results are not affected by these more computationally convenient choices. In appendix fig. S24 we show results for MNIST with $|S| = 10000$.

We use two DNN architectures: a relatively small vanilla two hidden-layer FCN with 784 inputs and 40 ReLU units in each hidden layer each, and also Resnet-50 (He et al., 2016), a 50-layer deep convolutional neural network, which is much closer to a state of the art (SOTA) system.

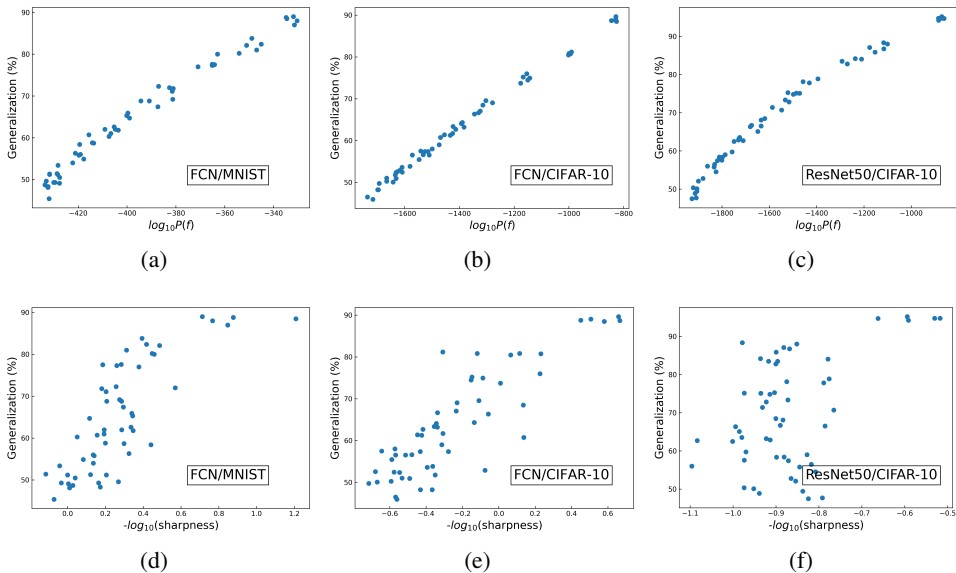

(a)          (b)          (c)

(d)          (e)          (f)

Figure 3: **The correlation between** $\log P(f)$**, sharpness and generalization accuracy on MNIST and CIFAR-10.** For MNIST $|S|$=500, $|E|$=1000; for CIFAR-10 $|S|$=5000, $|E|$=2000. The attack set size $|A|$ varies from 0 to $|S|$ and generates functions with different generalization performance. (a)-(c) depicts the correlation between generalization and $\log P(f)$ for FCN on MNIST, FCN on CIFAR-10 and Resnet-50 on CIFAR-10, respectively. (d)-(f) show the correlation between generalization and flatness for FCN on MNIST, FCN on CIFAR-10, and Resnet50 on CIFAR-10, respectively. In this experiment, all DNNs are trained with vanilla SGD.

We measure the flatness on cross-entropy (CE) loss at the epoch where SGD first obtains zero training error. Because the Hessian is so expensive to calculate, we mainly use the sharpness/flatness measure (definition 2.1) which is proportional to the Frobenius norm of the Hessian. The final error is measured in the standard way, after applying a sigmoid to the last layer to binarize the outputs.

To measure the prior, we use the GPs to which these networks reduce in the limit of infinite width (Lee et al., 2017; Matthews et al., 2018; Novak et al., 2018b). As demonstrated in Mingard et al. (2021a), GPs can be used to approximate the Bayesian posteriors $P_B(f|S)$ for finite width networks. For further details, we refer to the original papers above and to appendix D.

In order to generate functions $f$ with zero error on the training set $S$, but with diverse generalization performance, we use the attack-set trick from Wu et al. (2017). In addition to training on $S$, we add an attack set $A$ made up of incorrectly labelled data. We train on both $S$ and $A$, so that the error on $S$ is zero but the generalization performance on a test set $E$ is reduced. The larger $A$ is w.r.t. $S$, the worse the generalization performance. As can be seen in fig. 3(a)-(c), this process allows us to significantly vary the generalization performance. The correlation between $\log P(f)$ and generalization error is excellent over this range, as expected from our arguments in section 3.

Figs.3(d)-(f) show that the correlation between flatness and generalization is much more scattered than for $\log P(f)$. In appendix G we also show the direct correlation between $\log P(f)$ and flatness which closely resembles fig. 3(d)-(f) because $V(f)$ and $\epsilon$ correlate so tightly.

### 5.3 THE EFFECT OF OPTIMIZER CHOICE ON FLATNESS

We then test the effect of changing the optimizer from the vanilla SGD we used in fig. 3. We use Adam (Kingma & Ba, 2014), and entropy-SGD (Chaudhari et al., 2019) which includes an explicit term to maximize the flatness. Both SGD variants show good optimization performance for the standard default Tensorflow hyperparameters we use. Their generalization performance, however, does not significantly vary from plain SGD, and this is reflected in the priors of the functions that they find. More importantly, fig. 4 shows that the generalization-flatness correlation can be broken by

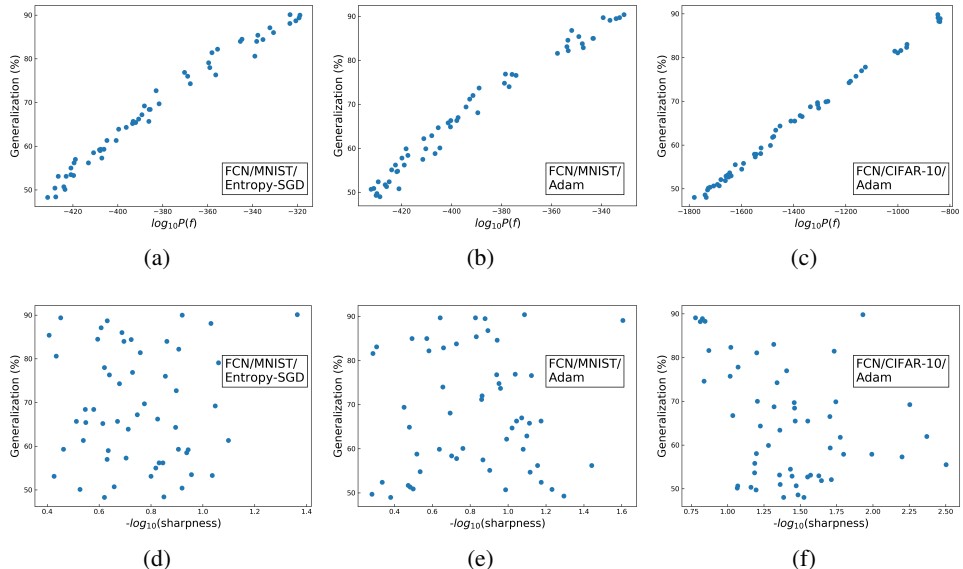

(a)  (b)  (c)

(d)  (e)  (f)

Figure 4: **SGD-variants can break the flatness-generalization correlation, but not the** $\log P(f)$**-generalization correlation.** The figures show generalization v.s. $\log P(f)$ or flatness for the FCN trained on (a) and (d) – MNIST with Entropy-SGD; (b) and (e) – MNIST with Adam; (c) and (f) – CIFAR-10 with Adam. for the same $S$ and $E$ as in fig. 3. Note that the correlation with the prior is virtually identical to vanilla SGD, but that the correlation with flatness measures changes significantly.

using these optimizers, whereas the $\log P(f)$-generalization correlation remains intact. A similar breakdown of the correlation persists upon overtraining and can also be seen for flatness measures that use Hessian eigenvalues (fig. S16 to fig. S21).

Changing optimizers or changing hyperparameters can, of course, alter the generalization performance by small amounts, which may be critically important in practical applications. Nevertheless, as demonstrated in Mingard et al. (2021a), the overall effect of hyperparameter or optimizer changes is usually quite small on these scales. The large differences in flatness generated simply by changing the optimizer suggests that flatness measures may not always reliably capture the effects of hyperparameter or optimizer changes. Note that we find less deterioration when comparing SGD to Adam for Resnet50 on CIFAR-10, (fig. S22). The exact nature of these effects remains subtle.

### 5.4 TEMPORAL BEHAVIOR OF SHARPNESS AND $\log P(f)$

In the experiments above, the flatness and $\log P(f)$ metrics are calculated at the epoch where the system first reaches $100\%$ training accuracy. In fig. 5, we measure the prior and the flatness for each epoch for our FCN, trained on MNIST (with no attack set). Zero training error is reached at epoch 140, and we overtrain for a further 1000 epochs. From initialization, both the sharpness measure from Definition 2.1, and $\log P(f)$ reduce until zero-training error is reached. Subsequently, $\log P(f)$ stays constant, but the CE loss continues to decrease, as expected for such classification problems. This leads to a reduction in the sharpness measure (greater flatness) even though the function, its prior, and the training error don't change. This demonstrates that flatness is a relative concept that depends, for example, on the duration of training. In figs. S16 and S17 we show for an FCN on MNIST that the quality of flatness-generalization correlations are largely unaffected by overtraining, for both SGD and Adam respectively, even though the absolute values of the sharpness change substantially.

One of the strong critiques of flatness is that re-parameterizations such as the parameter-rescaling transformation defined in eq. (1) can arbitrarily change local flatness measures (Dinh et al., 2017). Fig. 5 shows that parameter-rescaling indeed leads to a spike in the sharpness measure (a strong reduction in flatness). As demonstrated in the inset, the prior is initially invariant upon parameter-rescaling because $f(\mathbf{w})$ is unchanged. However, parameter-rescaling can drive the system to unusual

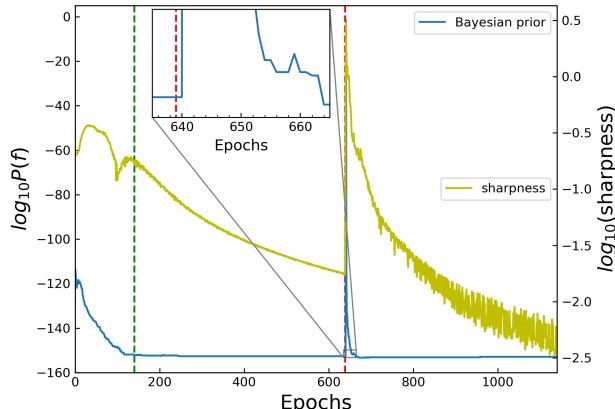

Figure 5: **How flatness evolves with epochs.** At each epoch we calculate the sharpness measure from Definition 2.1 (sharpness is the inverse of flatness) and the prior for our FCN on MNIST with $|S| = 500$. The green dashed line denotes epoch 140 where zero-training error is reached and post-training starts. The red dashed line denotes epoch 639 where parameter-rescaling takes place with $\alpha = 5.9$. Upon parameter-rescaling, the sharpness increases markedly, but then quickly decreases again. The inset shows that the prior is initially unchanged after parameter-rescaling. However, large gradients mean that in subsequent SGD steps, the function (and its prior) changes, before recovering to (nearly) the same function and $\log P(f)$.

parts of the volume with steep gradients in the loss function, which mean that SGD falls off the zero training error manifold. $\log P(f)$ goes up because it is more likely to randomly fall onto large $V(f)$ functions. However, the system soon relaxes to essentially the same function and $\log P(f)$. In fig. S11, we show that it is possible to obtain a spike in the sharpness measure without the prior changing. In each case, the sharpness measure rapidly decays after the spike, suggesting that parameter-rescaling brings the system into a parameter region that is "unnatural".

## 6 DISCUSSION AND FUTURE WORK

The notion that flatness correlates with generalization is widely believed in the community, but the evidential basis for this hypothesis has always been mixed. Here we performed extensive empirical work showing that flatness can indeed correlate with generalization. However, this correlation is not always tight, and can be easily broken by changing the optimizer, or by parameter-rescaling. By contrast, the $P(f)$ which is directly proportional to the Bayesian posterior $P_B(f|S)$ for functions that give zero error on the training set, is a much more robust predictor of generalization.

While the generalization performance of a DNN can be successfully predicted by the marginal likelihood PAC-Bayes bound (Valle-Pérez et al., 2018; Valle-Pérez & Louis, 2020), no such tight bound exists (to our knowledge) linking generalization and the Bayesian prior or posterior at the level of individual functions. Further theoretical work in this direction is needed. Moreover, it is natural to further extend current work towards linking flatness and the prior to other quantities which correlate with generalization such as frequency (Rahaman et al., 2018; Xu et al., 2019), or the sensitivity to changes in the inputs (Arpit et al., 2017; Novak et al., 2018a). Improvements to the GP approximations we use are an important technical goal. $P(f)$ can be expensive to calculate, so finding reliable local approximations related to flatness may still be a worthy endeavour. Finally, our main result – that $\log P(f)$ correlates so well with generalization – still requires a proper theoretical underpinning, notwithstanding the bound in eq.(4). Such explanations will need to include not just the networks and the algorithms, but also the data (Zdeborová, 2020). We refer readers to appendix A for more discussion on related works.

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

# A  MORE RELATED WORK

## A.1  PRELIMINARIES: TWO KINDS OF QUESTIONS GENERALIZATION AND TWO TYPES OF INDUCTIVE BIAS

In this supplementary section we expand on our briefer discussion of related work in the Introduction of the main paper. The question of why and how DNNs generalize in the overparameterized regime has generated a vast literature. To organize our discussion, we follow (Mingard et al., 2021a) and first distinguish two kinds of questions about generalization in overparameterized DNNs:

**1) The question of over-parameterized generalization**: Why do DNNs generalize at all in the overparameterized regime, where classical learning theory suggests they should heavily overfit.

**2) The question of fine-tuned generalization**: Given that a DNN already generalizes reasonably well, how can detailed architecture choice, optimizer choice, and hyperparameter tuning further improve generalization?

Question 2) is the main focus of a large tranche of the literature on generalization, and for good reason. In order to build SOTA DNNs, even a few percent accuracy improvement (taking image classification as an example) is important in practice. Improved generalization performance can be achieved in many ways, including local adjustments of the DNNs structure (e.g. convolutional layers, pooling layers, shortcut connections etc.), hyperparameter tuning (learning rate, batch size etc.), or choosing different optimizers (e.g. vanilla SGD versus entropySGD (Chaudhari et al., 2019) or Adam (Kingma & Ba, 2014). In this paper, however, we are primarily interested in question 1). As pointed out, for example famously in Zhang et al. (2016), but also by many researchers before that [1], DNNs can be proven to be highly expressive, so that the number of hypotheses that can fit a training data set $S$, but generalize poorly, is typically many orders of magnitude larger than the number that can actually generalize. And yet in practice DNNs do not tend to overfit much, and can generalize well, which implies that DNNs must have some kind of *inductive bias* (Shalev-Shwartz & Ben-David, 2014) toward hypotheses that generalize well on unseen data.

Following the framework of (Mingard et al., 2021a), we use the language of functions (rather than that of hypotheses, see also appendix B.) to distinguish two major potential types of inductive bias.

**A) The inductive bias upon upon random sampling of parameters over a parameter distribution $P_w(\mathbf{w})$.** In other words, given a DNN architecture, loss function etc. and a measure over parameters $P_w(\mathbf{w})$ (which can be taken to be the initial parameter distribution for an optimizer, but is more general), this bias occurs when certain types of functions more likely to appear upon random sampling of parameters than others. This inductive bias can be expressed in terms of a prior over functions $P(f)$, or in terms of a posterior $P_B(f|S)$ when the functions are conditioned, for example, on obtaining zero error on training set $S$.

**B) The inductive bias induced by optimizers during a training procedure.** In other words, given an inductive bias upon initialization (from **A**), does the training procedure induce a further inductive bias on what functions a DNN expresses? One way of measuring this second form of inductive bias is to calculate the probability $P_{opt}(f|S)$ that an DNN trained to zero error on training set $S$ with optimizer *opt* (typically a variant of SGD) expresses function $f$, and then compare it to the Bayesian posterior probability $P_B(f|S)$ that this function obtains upon random sampling of parameters (Mingard et al., 2021a). In principle $P_B(f|S)$ expresses the inductive bias of type A), so any differences between $P_{opt}(f|S)$ and $P_B(f|S)$ could be due to inductive biases of type B).

These two sources of inductive bias can be relevant to both questions above about generalization. We emphasize that our taxonomy of two questions about generalization, and two types of inductive bias is just one way of parsing these issues. We make these first order distinctions to help clarify our discussion of the literature, and are aware that there are other ways of teasing out these distinctions.

## A.2  RELATED WORK ON FLATNESS

The concept "flatness" of the loss function of DNNs can be traced back to Hinton & van Camp (1993) and Hochreiter & Schmidhuber (1997). Although these authors did not provide a completely formal mathematical definition of flatness, Hochreiter & Schmidhuber (1997) described flat minima as "a large connected region in parameter space where the loss remains approximately constant", which requires lower precision to specify than sharp minima. They linked this idea to the minimum description length (MDL) principle (Rissanen, 1978), which says that the best performing model is the one with shortest description length, to argue that flatter minima should generalize better than sharp minima. More generally, flatness can be interpreted as a complexity control of the hypotheses class introduced by algorithmic choices.

---

[1]For example, Leo Breiman, included the question of overparameterised generalization in DNN back in in 1995 as one of the main issues raised by his reflections on 20 years of refereeing for NEURIPS (Breiman, 1995)).

The first thing to note is that flatness is a property of the functions that a DNN converges on. In other words, the basic argument above is that flatter functions will generalize better, which can be relevant to both questions 1) and 2) above.

It is a different question to ask whether a certain way of finding functions (say by optimizing a DNN to zero error on a training set) will generate an inductive bias towards flatter functions. In Hochreiter & Schmidhuber (1997), the authors proposed an algorithm to bias towards flatter minima by minimizing the training loss while maximizing the log volume of a connected region of the parameter space. This idea is similar to the recent suggestion of entropy-SGD (Chaudhari et al., 2019), where the authors also introduced an extra regularization to bias the optimizer into wider valleys by maximizing the "local entropy".

In an influential paper, Keskar et al. (2016) reported that the solutions found by SGD with small batch sizes generalize better than those found with larger batch sizes, and showed that this behaviour correlated with a measure of "sharpness" (sensitivity of the training loss to perturbations in the parameters). Sharpness can be viewed as a measure which is the inverse of the flatness introduced by Hinton & van Camp (1993) and Hochreiter & Schmidhuber (1997). This work helped to popularize the notion that SGD itself plays an important role in providing inductive bias, since differences in generalization performance and in sharpness correlated with batch size. In follow-on papers others have showed that the correlation with batch size is more complex, as some of the improvements can be mimicked by changing learning rates or number of optimization steps for example, see (Hoffer et al., 2017; Goyal et al., 2017; Smith et al., 2017; Neyshabur et al., 2017b). Nevertheless, these changes in generalization as a function of optimizer hyperparameters are important things to understand because they are fundamentally type B) inductive bias. Because the changes in generalization performance in these papers tend to be relatively small, they mainly impinge on question 2) for fine-tuned generalization. Whether these observed effects are relevant for question 1) is unclear from this literature.

Another strand of work on flatness has been through the lens of generalization bounds. For example, Neyshabur et al. (2017b) showed that sharpness by itself is not sufficient for ensuring generalization, but can be combined, through PAC-Bayes analysis, with the norm of the weights to obtain an appropriate complexity measure. The connection between sharpness and the PAC-Bayes framework was also investigated by Dziugaite & Roy (2017), who numerically optimized the overall PAC-Bayes generalization bound over a series of multivariate Gaussian distributions (different choices of perturbations and priors) which describe the KL-divergence term appearing in the second term in the combined generalization bound by Neyshabur et al. (2017b). For more discussion of this literature on bounds and flatness, see also the recent review (Valle-Pérez & Louis, 2020). Rahaman et al. (2018) also draw a connection to flatness through the lens of Fourier analysis, showing that DNNs typically learn low frequency components faster than high frequency components. This frequency argument is related to the input-output sensitivity picture, which is systematically investigated in Novak et al. (2018a).

There is also another wide-spread belief that SGD trained DNNs are implicitly biased towards having small parameters norms or large margin, intuitively inspired by classical ridge regression and SVMs. Bartlett et al. (2017) presented a margin-based generalization bound that depends on spectral and $L_{2,1}$ norm of the layer-wise weight matrices of DNNs. Neyshabur et al. (2017a) later proved a similar spectral-normalized margin bound using PAC-Bayesian approach rather than the complex covering number argument used in Bartlett et al. (2017). Liao et al. (2018) further strengthen the theoretical arguments that an appropriate measure of complexity for DNNs should be based on a product norm by showing the linear relationship between training/testing CE loss of normalized networks. Jiang et al. (2018) also empirically studied the role of margin bounds.

In a recent important large-scale empirical work on different complexity measures by Jiang et al. (2019), 40 different complexity measures are tested when varying 7 different hyperparameter types over two image classification datasets. They do not introduce random labels so that data complexity is not thoroughly investigated. Among these measures, the authors found that sharpness-based measures outperform their peers, and in particular outperform norm-based measures. It is worth noting that their definition of "worst case" sharpness is similar to definition 2.1 but normalized by weights, so they are not directly comparable. In fact, their definition of worst case sharpness in the PAC-Bayes picture is more close to the works by Petzka et al. (2019); Rangamani et al. (2019); Tsuzuku et al. (2019) which focus on finding scale-invariant flatness measure. Indeed enhanced performance are reported in these works. However, these measures are only scale-invariant when the scaling is layer-wise. Other methods of re-scaling (e.g. neuron-wise re-scaling) can still change the metrics. Moreover, the scope of Jiang et al. (2019) is concentrated on the practical side (e.g. inductive bias of type B) and does not consider data complexity, which we believe is a key ingredient to understanding the inductive bias needed to explain question 1) on generalization.

Finally, in another influential paper, Dinh et al. (2017) showed that many measures of flatness, including the sharpness used in Keskar et al. (2016), can be made to vary arbitrarily by re-scale parameters while keeping the function unchanged. This work has called into question the use of local flatness measures as reliable guides to generalization, and stimulated a lot of follow on studies, including the present paper where we explicitly study how parameter-rescaling affects measures of flatness as a function of epochs.

### A.3 RELATED WORK ON THE INFINITE-WIDTH LIMIT

A series of important recent extensions of the seminal proof in Neal (1994) - that a single-layer DNN with random iid weights is equivalent to a GP (Mackay, 1998) in the infinite-width limit - to multiple layers and architectures (NNGPs) have recently appeared (Lee et al., 2017; Matthews et al., 2018; Novak et al., 2018b; Garriga-Alonso et al., 2019; Yang, 2019). These studies on NNGPs have used this correspondence to effectively perform a very good approximation to exact Bayesian inference in DNNs. When they have compared NNGPs to SGD-trained DNNs the generalization performances have generally shown a remarkably close agreement. These facts require rethinking the role SGD plays in question 1) about generalization, given that NNGPs can already generalize remarkably well without SGD at all.

### A.4 RELATIONSHIP TO PREVIOUS PAPERS USING THE FUNCTION PICTURE

The work in this paper builds on a series of recent papers that have explored the function based picture in random neural networks. We briefly review these works to clarify their connection to the current paper.

Firstly, in Valle-Pérez et al. (2018), the authors demonstrated empirically that upon random sampling of parameters, DNNs are highly biased towards functions with low complexity. This behaviour does not depend very much on $P_w(\mathbf{w})$ for a range of initial distributions typically used in the literature. Note that this behaviour does start to deviate from what was found in (Valle-Pérez et al., 2018), when the system enters a chaotic phase, which can be reached with for tanh or erf non-linearities and for $P_w(\mathbf{w})$ with a relatively large variance (Yang & Salman, 2019). They show more specifically that the bias towards simple functions is consistent with the "simplicity bias" from Dingle et al. (2018; 2020), which was inspired by the coding theorem from AIT (Li & Vitanyi, 2008), first derived by Levin (1974) . The idea of simplicity bias in DNNs states that if the parameter-function map is sufficiently biased, then the probability of the DNN producing a function $f$ on input data drops exponentially with increasing Kolmogorov complexity $K(f)$ of the function $f$. In other words, high $P(f)$ functions have low $K(f)$, and high $K(f)$ functions have low $P(f)$. A key insight from (Dingle et al., 2018; 2020) is that $K(f)$ can be approximated by an appropriate measure $\tilde{K}(f)$ and still be used to make predictions on $P(f)$, even if the true $K(f)$ is formally incomputable. Recently Mingard et al. (2019) and De Palma et al. (2018) gave two separate non-AIT based theoretical justifications for the existence of simplicity bias in DNNs. In other words, this line of work suggests that DNNs have an intrinsic bias towards simple functions upon random sampling of parameters, and in our taxonomy, that is bias of type A).

If simplicity bias in DNNs matches "natural" data distributions, then, at least upon random sampling of parameters, this should help facilitate good generalization. Indeed, it has been shown that data such as MNIST or CIFAR-10 is relatively simple (Lin et al., 2017; Goldt et al., 2019; Spigler et al., 2019), suggesting that an inductive bias toward simplicity will assist with good generalization.

A second paper upon which the current one builds is (Mingard et al., 2021a), where extensive empirical test (for a range of architectures (FCN, CNN, LSTM), datasets (MNIST, Fashion-MNIST, CIFAR-10, ionosphere, IMDb moviereview dataset), and SGD variants (vanilla SGD, Adam, Adagrad, RMSprop, Adadelta), as well as for different batch sizes and learning rates) were done of the hypothesis that:

$$P_{opt}(f|S) \approx P_B(f|S). \tag{5}$$

Here $P_{opt}(f|S)$ is the probability that an optimizer (SGD or one of its variants) converges upon a function $f$ after training to zero training error on a training set $S$. By training over many different parameter initializations, $P_{opt}(f|S)$ can be calculated. Similarly, the Bayesian posterior probability $P_B(f|S)$ is defined as the probability that upon random sampling of parameters, a DNN expresses function $f$, conditioned on zero error on $S$. The functions were, as in the current paper, a restriction to a given training set $S$ and test set $E$. Since the systems always had zero error on the training set, functions could be compared by what they produced on the test set (for example, the set of labels on the images for image classification). It was found that the hypothesis (A.4) held remarkably well to first order, for a wide range of systems. At first sight this similarity is surprising, given that the procedures to generate $P_{opt}(f|S)$ (training with an optimizer such as SGD) is completely different from those for $P_B(f|S)$ (where GP techniques and direct sampling were used), which knows nothing of optimizers at all. The fact that these two probabilities are so similar suggests that any inductive bias of type B), which would be a bias beyond what is already present in $P_B(f|S)$, is relatively small. While this conclusion does not imply that there are no induced biases of type B), and clearly there are since hyperparameter tuning affects fine-tuned generalization, it does suggest that the main source of inductive bias needed to explain 1), the question of why DNNs generalize in the first place, is found in the inductive biases of type A), which are already there in $P_B(f|S)$. In (Mingard et al., 2021a), the authors propose that, for highly biased priors $P(f)$, that SGD is dominated by the large differences in basin size for the different functions $f$, and so finds functions with probabilities dominated by the initial distribution. A similar effect was seen in evolutionary systems (Schaper & Louis, 2014; Dingle et al., 2015) where it was called the arrival of the frequent.

In addition, in (Mingard et al., 2021a), the authors observed for one system that $-\log(P_B(f|S))$ scaled linearly with the generalization error on $E$ for a wide range of errors. This preliminary result provided inspiration for the current paper where we directly study the correlation between the prior $P(f)$ and the generalization error.

The third main function based paper that we build upon is (Valle-Pérez & Louis, 2020) which provides a comprehensive analysis of generalization bounds. In particular, it studies in some detail the Marginal Likelihood PAC-Bayes bound, first presented in Valle-Pérez et al. (2018), which is predicts a direct link between the generalization error and the log of the marginal likelihood $P(S)$. $P(S)$ can be interpreted as the total prior probability that a function is found with zero error on the training set $S$, upon random sampling of parameters of the DNN. The performance of the bound was tested for challenges such as varying amounts of data complexity, different kinds of architectures, and different amounts of training data (learning curves). For each challenge it works remarkably well, and to our knowledge no other bound has been tested this comprehensively. Again, the good performance of this bound, which is agnostic about optimizers, suggest that a large part of the answer to question 1) can be found in the inductive bias of type A), e.g. that found upon initialization. The bound is not accurate enough to explain smaller effects relevant for fine-tuning generalization, which can originate from other sources such as a difference in optimizer hyperparameters. These conclusions are consistent with the different approach in this paper, where we use the prior $P(f)$ (which knows nothing about SGD) and show that it also correlates with predicted test error for DNNS trained with SGD and its variants. We do propose a simpler bound that is consistent with the observed scaling, but more work is needed to get anywhere near the rigour found in (Valle-Pérez & Louis, 2020) for the full marginal likelihood bound.

Finally, we note that in all three of these papers, GPs are used to calculate marginal likelihoods, posteriors, and priors. Technical details of how to use GPs can be found clearly explained there.

The current paper *builds* on this body of work and uses some of the techniques described therein, but it is distinct. Firstly, our measurements on flatness are new, and our claim that the prior $P(f)$ correlates with generalization, while indirectly present in (Mingard et al., 2021a) was not developed there at all as that paper focuses on the posterior $P_B(f|S)$, and did not use the attack set trick to vary functions that are consistent with $S$, and so is tackling a different question (namely how much extra inductive bias comes from using SGD over the inductive bias already present in the Bayesian posterior). The attack set trick means that $P(S)$ does not change, while clearly the generalization error (or expected test error) does change, so the marginal likelihood bound is not predictive here.

## B PARAMETER-FUNCTION MAP AND NEUTRAL SPACE

The link between the parameters of a DNN and the function it expresses is formally described by the parameter-function map:

**Definition B.1** (Parameter-function map)**.** Consider the model defined in definition 2.2, if the model takes parameters within a set $W \subseteq \mathbb{R}^n$, then the parameter-function map $\mathcal{M}$ is defined as

$$\mathcal{M} : W \to \mathcal{F}$$
$$\mathbf{w} \mapsto f_{\mathbf{w}}.$$

where $f_{\mathbf{w}}$ denotes the function parameterized by $\mathbf{w}$.

The parameter-function map, introduced in (Valle-Pérez et al., 2018), serves as a bridge between a parameter searching algorithm (e.g. SGD) and the behaviour of a DNN in function space. In this context we can also define the:

**Definition B.2** (Neutral space)**.** For a model defined in Definition B.1, and a given function $f$, the neutral space $\mathcal{N}_f \subseteq W$ is defined as

$$\mathcal{N}_f := \{\mathbf{w} \in W : \mathcal{M}(\mathbf{w}) = f\}.$$

The nomenclature comes from genotype-phenotype maps in the evolutionary literature (Manrubia et al., 2020), where the space is typically discrete, and a neutral set refers to all genotypes that map to the same phenotype. In this context, the Bayesian prior $P(f)$ can be interpreted as the probabilistic volume of the corresponding neutral space.

## C CLARIFICATION ON DEFINITION OF FUNCTIONS AND PRIOR

The discussion of "functions" represented by DNNs can be confusing without careful definition. In fig. S6 we list four different interpretations of "functions" commonly seen in literature which also are directly related to our work. These interpretations cover both regression and classification settings. Let $\mathcal{X}$ be an arbitrary input domain and $\mathcal{Y}$ be the output space. According to different interpretations of the function represented by a DNN, $\mathcal{Y}$ will be different, for the same choice of $\mathcal{X}$ and DNN.

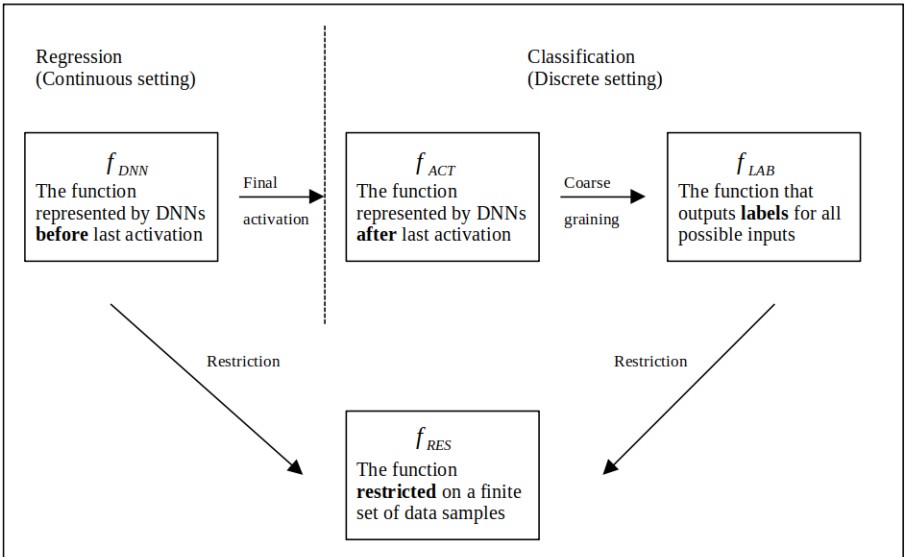

Figure S6: The diagram of different definitions for functions represented by DNNs.

**Definition C.1** ($f_{\text{DNN}}$). Consider a DNN whose input domain is $\mathcal{X}$. Then $f_{\text{DNN}}$ belongs to a class of functions $\mathcal{F}_{\text{DNN}}$ which define the mapping between $\mathcal{X}$ to the pre-activation of the last layer of DNN, which lives in $\mathbb{R}^d$:

$$f_{\text{DNN}} \in \mathcal{F}_{\text{DNN}} : \mathcal{X} \to \mathbb{R}^d$$

$d$ is the width of the last layer of DNN.

In standard GP terminology, $f_{\text{DNN}}$ is also called *latent function* (Rasmussen, 2003). This is the function we care about in regression problems.

In the context of supervised learning, we have to make some assumptions about the characteristics of $\mathcal{F}_{\text{DNN}}$, as otherwise we would not know how to choose between functions which are all consistent with the training sample but might have hugely different generalization ability. This kind of assumptions are called *inductive bias*. One common approach of describing the inductive bias is to give a prior probability distribution to $\mathcal{F}_{\text{DNN}}$, where higher probabilities are given to functions that we consider to be more likely. For DNNs, $\mathcal{F}_{\text{DNN}}$ is a set of functions over an (in general) uncountably infinite domain $\mathcal{X}$. There are several approaches to define probability distributions over such sets. GP represent one approach, which generalizes Gaussian distributions to function spaces. If we ask only for the properties of the functions at a finite number of points, i.e. restriction of $\mathcal{F}_{\text{DNN}}$ to $C : \{c_1, \ldots, c_m\} \subset \mathcal{X}$ (see definition 2.2), then inference with a GP, reduces to inference with a standard multidimensional Gaussian distribution. This is an important property of GP called *consistency*, which helps in making computations with GP feasible. As shown in appendix D, we can readily compute with this GP prior over $\mathcal{F}_{\text{DNN}}$ as long as it is restricted on a finite data set. Later in definition C.4 we will formally define the restricted function $f_{\text{RES}}$.

In classification tasks, we typically get a data sample from $\mathcal{X} \times \mathcal{Y}$, where without loss of generality $\mathcal{Y}$ has the form of $\mathcal{Y} = \{1, \ldots, k\}$ where $k$ is the number of classes. For simplicity, we further assume binary classification where $\mathcal{Y} = \{0, 1\}$ Note in the scope of binary classification we have the last layer width of $d = 1$. To grant the outputs of the function represented by a DNN a probability interpretation, we need the outputs lie in the interval $(0, 1)$. One way of doing so is to "squash" the outputs of $f_{\text{DNN}}$ to $(0, 1)$ by using a final *activation*, typically a logistic or sigmoid function $\lambda(z) = (1 + \exp(-z))^{-1}$. Subsequently we have the definition of $f_{\text{ACT}}$ in fig. S6:

**Definition C.2** ($f_{\text{ACT}}$). Consider the setting and $f_{\text{DNN}}$ defined in definition C.1 where $d = 1$, and a logistic activation $\lambda(z) = (1 + \exp(-z))^{-1}$. Then $f_{\text{ACT}}$ is defined as :

$$f_{\text{ACT}} := f_{\text{DNN}} \circ \lambda : \mathcal{X} \to (0, 1)$$

where $\circ$ denotes function composition. we also define the space of $f_{\text{ACT}}$ as

$$\mathcal{F}_{\text{ACT}} = \{f_{\text{ACT}} \text{ for every } f_{\text{DNN}} \in \mathcal{F}_{\text{DNN}}\}$$

In real life classification datasets, we typically do not have access to the probability of an input classified as one certain label, but the labels instead. When we discuss functions represented by DNNs in classification, we

usually mean the *coarse-grained* version of $f_{\mathrm{ACT}} \in \mathcal{F}_{\mathrm{ACT}}$, meaning we group all outputs to 1 if the probability of predicting the inputs as being label "1" is greater or equal than 0.5, and 0 otherwise. Mathematically, we define $f_{\mathrm{LAB}}$ as:

**Definition C.3** ($f_{\mathrm{LAB}}$)**.** Consider the setting and $f_{\mathrm{ACT}}$ defined in definition C.2 and a threshold function

$$\tau(z) = \begin{cases} 1 \text{ if } z \geq 0.5 \\ 0 \text{ otherwise .} \end{cases}$$

Then we define $f_{\mathrm{LAB}}$ and the space $\mathcal{F}_{\mathrm{LAB}}$ as:

$$f_{\mathrm{LAB}} = f_{\mathrm{ACT}} \circ \tau : \mathcal{X} \to \{0, 1\}$$

$$\mathcal{F}_{\mathrm{LAB}} = \{f_{\mathrm{LAB}} \text{ for every } f_{\mathrm{ACT}} \in \mathcal{F}_{\mathrm{ACT}}\}$$

The definition C.3 allows us to describe the function represented by a DNN in binary classification as a binary string consisting of "0" and "1", whose length is equal to the size of input domain set $|\mathcal{X}|$. As explained earlier, in classification we also want to put a prior over $\mathcal{F}_{\mathrm{LAB}}$ and use this prior as our belief about the task before seeing any data.

Finally, as we mentioned above, to make computations tractable, we restrict the domain to a finite set of inputs. We use the definition of restriction in definition 2.2 to formally define the "functions" we mean and practically use in our paper:

**Definition C.4** ($f_{\mathrm{RES}}$)**.** Consider a DNN whose input domain is $\mathcal{X}$ with a last layer width $d = 1$ . Let $C = \{c_1, \ldots, c_m\} \subset \mathcal{X}$ be any finite subset of $\mathcal{X}$ with cardinality $m \in \mathbb{N}$. The restriction of function space $\mathcal{F} \in \{\mathcal{F}_{\mathrm{DNN}}, \mathcal{F}_{\mathrm{LAB}}\}$ to $C$ is denoted as $\mathcal{F}^C$, and is defined as the space of all functions from $C$ to $\mathcal{Y}$ realizable by functions in $\mathcal{F}$. We denote with $f_{\mathrm{RES}}$ elements of their corresponding spaces of restricted functions. Specifically, in regression:

$$f_{\mathrm{RES}} \in \mathcal{F}_{\mathrm{DNN}}^C : C \to \mathbb{R}$$

and in binary classification:

$$f_{\mathrm{RES}} \in \mathcal{F}_{\mathrm{LAB}}^C : C \to \{0, 1\}$$

Note that in definition C.4 we only consider scalar outputs in the regression setting. For multiple-output functions, one approach is to consider $d$ GPs and compute the combined kernel (Alvarez et al., 2011).

In statistical learning theory, the function spaces $\mathcal{F}_{\mathrm{DNN}}$ and $\mathcal{F}_{\mathrm{LAB}}$ are also called *hypotheses classes*, with their elements called *hypotheses* (Shalev-Shwartz & Ben-David, 2014). It is important to note that our definition of prior and its calculation is based on the restriction of the hypotheses class to the concatenation of training set and test set $S + E$. Mathematically, this means the prior of a function $P(f)$ we calculated in the paper is precisely $P(f_{\mathrm{RES}})$, except for the Boolean system in section 5.1, where the input domain $\mathcal{X}$ is discrete and small enough to enumerate (this can also be thought of as the trivial restriction). As explained above, this restriction is inevitable if we want to compute the prior over $\mathcal{F}_{\mathrm{DNN}}$ or $\mathcal{F}_{\mathrm{LAB}}$. A simple example on MNIST (LeCun et al., 1998) can also help to gain a intuition of the necessity of such restriction, where all inputs would include the set of 28x28 integer matrices whose entries take values from 0-255, which gives $256^{784}$ possible inputs. This indicates that for real-life data distributions the number of all possible inputs is hyper-astronomically large, if not infinite. Nevertheless, In some cases, such as the Boolean system described in Valle-Pérez et al. (2018) and treated in section 5.1, there is no need for such restriction because it is feasible to enumerate all possible inputs: there are only 7 Boolean units which give $2^7 = 128$ possible data sample. However, even in such cases, the number of possible functions is still large ($2^{128} \approx 10^{38}$).

# D  GP APPROXIMATION OF THE PRIOR

In this section, we sketch out how we calculated the prior of a function $P(f)$ (Valle-Pérez et al., 2018; Mingard et al., 2021a). As in those papers, we use GP, which have been shown to be equivalent to DNNs in the limit of infinite layer width (Neal, 1994; Lee et al., 2017; Matthews et al., 2018; Tan, 2008; Rasmussen, 2003). These neural network GPs (NNGPs) have been shown to accurately approximate the prior over functions $P(f)$ of finite-width Bayesian DNNs (Valle-Pérez et al., 2018; Matthews et al., 2018; Mingard et al., 2021a).

For the NNGPs, a GP prior is placed on the pre-activations $z$ of the last layer of the neural network (before a final non-linearity, e.g. softmax, is applied), meaning that for any finite inputs set $\mathbf{x} = \{x_1, \ldots, x_m\}$, the random output vector (pre-activations) $\boldsymbol{z} = [z(x_1), \ldots, z(x_m)]^T$ has a Gaussian distribution. Note that in this paper, the the last layer has a single activation since we only focus on binary classification. This setting is corresponding to the definition of function restriction is definition C.4, with $\boldsymbol{z} \in \mathbb{R}^m$. Without loss of generality, we can assume such a process has a zero mean. The prior probability of the outputs $\boldsymbol{z}$ can be calculated as:

$$P(\boldsymbol{z}) = \frac{1}{(2\pi)^{\frac{m}{2}} \Sigma^{\frac{1}{2}}} \exp\left(-\frac{1}{2}\boldsymbol{z}^T \Sigma^{-1} \boldsymbol{z}\right) \tag{6}$$

$\Sigma$ is the covariance matrix (often called kernel), whose entries are defined as $\Sigma(x_i, x_j) \equiv \mathbb{E}[z(x_i), z(x_j)]$. Neal (1994) gave the basic form of kernel $\Sigma$ in single hidden layer case, where $\Sigma$ depends on the variance of weights and biases ($\sigma_w$ and $\sigma_b$). In DNNs with multiple hidden layers, the kernel for layer $l$ can be calculated recursively by induction, assuming the layer $l-1$ is a GP (Lee et al., 2017; Matthews et al., 2018). The kernel for fully connected ReLU-activated networks has a well known *arc-cosine kernel* analytical form (Cho & Saul, 2009), which we used in all FCNs in our work.

For ResNet50, the analytical form of GP kernel is intractable. Instead, we use a Monte Carlo empirical kernel (Novak et al., 2018b), and apply one step of the fully connected GP recurrence relation (Lee et al., 2017), taking advantage of the fact that the last layer of ResNet50 is fully connected. Mathematically, the empirical kernel can be expressed as:

$$\tilde{\Sigma}(x_i, x_j) := \frac{\sigma_w^2}{Mn} \sum_{m=1}^{M} \sum_{c=1}^{n} \left( h_{\mathbf{w}_m}^{L-1}(x_i) \right)_c \left( h_{\mathbf{w}_m}^{L-1}(x_j) \right)_c + \sigma_b^2 \tag{7}$$

where $\left( h_{\mathbf{w}_m}^{L-1}(x) \right)_c$ is the activation of $c$-th neuron in the last hidden layer ($L$ is the total number of layers) for the network parameterized by the $m$-th sampling of parameters $\mathbf{w}_m$, $M$ is the number of total Monte Carlo sampling, $n$ is the width of the final hidden layer, and $\sigma_w$, $\sigma_b$ are the weights and biases variance respectively. In our experiments, $M$ is set to be $0.1 \times (|S| + |E|)$.

After calculating $P(\mathbf{z})$ with the corresponding kernel, the prior over (coarse-grained) restriction of functions $P(f)$ can be calculated through likelihood $P(f|\mathbf{z})$, which in our case is just a Heaviside function representing a hard sign nonlinearity. As non-Gaussian likelihood produces an intractable $P(f)$, we used Expectation Propagation (EP) algorithm for the approximation of $P(f)$ (Rasmussen, 2003). This same EP approximation was used in Mingard et al. (2021a) where it is discussed further. We represent the function $f$ by the input-output pairs on the concatenation of training set and test set $S + E$.

## E    COMPARING FLATNESS METRICS

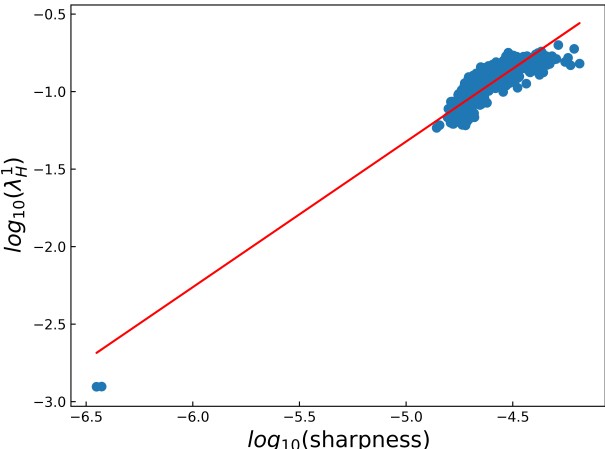

Figure S7: The direct correlation between sharpness and spectral norm of Hessian for the 1000 most frequently found functions found after SGD runs for a two hidden layer FCN, in the $\mathbf{n} = \mathbf{7}$ Boolean system (Same system as in fig. 2) .

As mentioned in section 2.2 of the main text, the sharpness metric in definition 2.1 can be directly linked to spectral norm of the Hessian by considering the second order Taylor expansion of $L(\mathbf{w})$ around a critical point in powers of $\zeta$ (Dinh et al., 2017). We empirically confirm this relationship by showing in fig. S7 the direct correlation between sharpness and spectral norm of Hessian, as well as in fig. S8 the correlation between Hessian spectral norm and prior in Boolean system described in section 5.1.

In addition to the spectral norm, another widely used flatness measure is the product of a subset of the positive Hessian eigenvalues, typically say the product of the top-50 largest eigenvalues (Wu et al., 2017; Zhang et al., 2018). We measured the correlation of these Hessian-based flatness metrics with sharpness as well as with generalization for the FCN/MNIST system in fig. S9. Since they correlate well with the sharpness, these flatness measures show very similar correlations with generalization as sharpness does in fig. 3 and fig. 4. In other words, the Hessian-based flatness metrics also capture the loose correlation with generalization when the neural network is trained by SGD and the deterioration of this correlation when we change the optimizer to Adam.

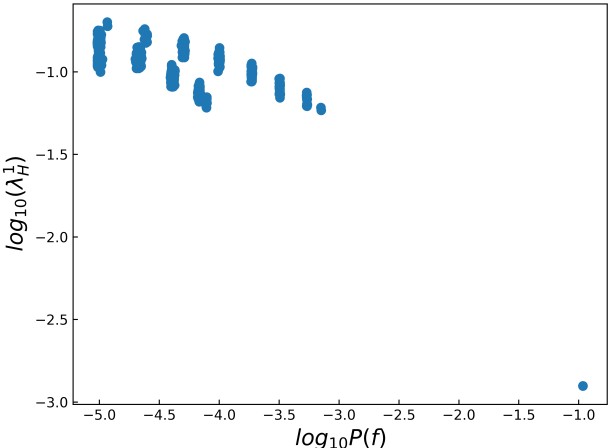

Figure S8: The correlation between prior and flatness in Boolean system where the flatness is measured by spectral norm of Hessian, for the 1000 most frequently occurring functions found by SGD runs with a two hidden layer FCN. The system is the same **n = 7** Boolean system as in fig. 2 except that we use a different metric of flatness.

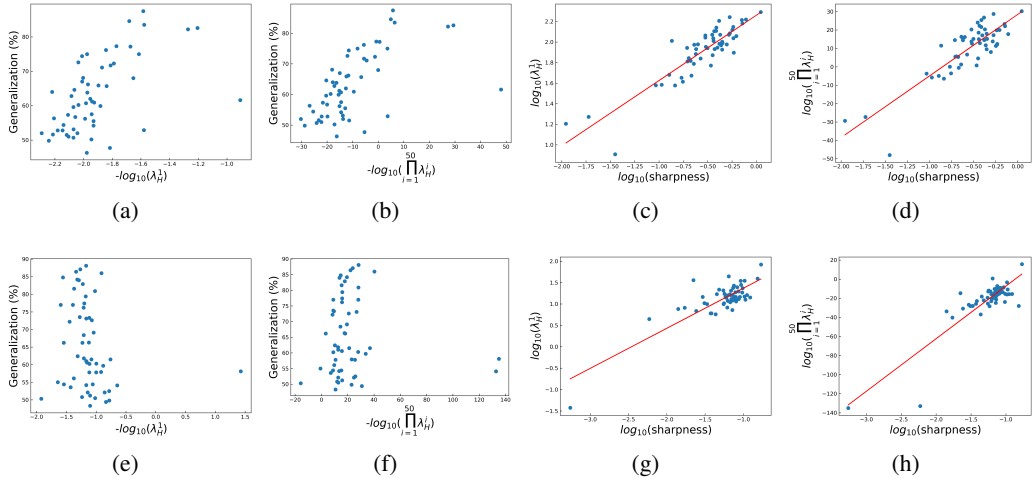

Figure S9: Two Hessian-based flatness metrics show analogous behavior to sharpness defined in (definition 2.1). The architecture and dataset are FCN/MNIST, with training set size $|S| = 500$, and test set size $|E| = 1000$; which are the same settings as fig. 3 (d) and fig. 4 (e). **Optimizer: SGD** (a) - (b): The correlation between Hessian-based flatness metrics and generalization. (c) - (d): Sharpness and Hessian-based flatness metrics correlate well with one another. **Optimizer: Adam** (e) - (f): The correlation between Hessian-based flatness metrics and generalization breaks down, just as it does for sharpness in fig. 4. (g) - (h): Sharpness and Hessian-based flatness metrics correlate well with one another, even though they don't correlate well with generalization.

Another detail worth noting is that Keskar et al. (2016) used the L-BFGS-B algorithm (Byrd et al., 1995) to perform the maximization of $L(\mathbf{w})$ in $\mathcal{C}_\zeta$, which is the box boundary around the minimum of interest:

$$\mathcal{C}_\zeta = \{\Delta\mathbf{w} \in \mathbb{R}^n : -\zeta\left(|w_i| + 1\right) \leq \Delta w_i \leq \zeta\left(|w_i| + 1\right) \quad \forall i \in \{1, 2, \cdots, n\}\} \tag{8}$$

However, as a quasi-Newton method, L-BFGS-B is not scalable when there are tens of millions of parameters in modern DNNs. To make Keskar-sharpness applicable for large DNNs (e.g. ResNet50), we use vanilla SGD for the maximization instead. The hyperparameters for the sharpness calculation are listed in table 1. Note that the entries batch size, learning rate and number of epochs all refer to the SGD optimizer which does the maximization in the sharpness calculation process. The number of epochs is chosen such that the max value of

Table 1: Hyperparameters for sharpness calculation

| DATA SET | ARCHITECTURE | BOX SIZE ($\zeta$) | BATCH SIZE | LEARNING RATE | NUMBER OF EPOCHS |
|---|---|---|---|---|---|
| BOOLEAN | FCN | $10^{-4}$ | 16 | $10^{-3}$ | 10 |
| MNIST | FCN | $10^{-4}$ | 32 | $10^{-3}$ | 100 |
| CIFAR10 | FCN | $10^{-5}$ | 128 | $5 \times 10^{-5}$ | 100 |
| CIFAR10 | RESNET50 | $10^{-5}$ | 128 | $10^{0}$ | 100 |

loss function found at each maximization step converges. An example of the convergence of sharpness is shown in fig. S10. As a check, we also compared our SGD-sharpness with the original L-BFGS-B-sharpness, finding similar results.

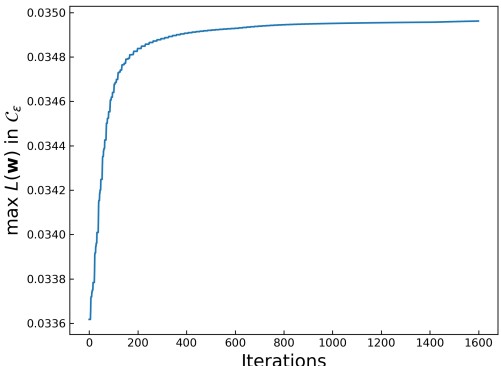

Figure S10: The max value of loss function $L(\mathbf{w})$ at each iteration in the process of maximization, when calculating the sharpness using SGD instead of L-BFGS-B. The plot shows the max loss value found by SGD in the box limit $\mathcal{C}_\zeta$ will converge after given number of epochs. For this plot the hyperparameters are listed in the second line of table 1 (MNIST).

## F    IMPLEMENTING PARAMETER RE-SCALING

In this section we describe in detail how we implement the alpha scaling in DNNs first proposed by Dinh et al. (2017). The widely used rectified linear activation (ReLU) function

$$\phi_{\text{rect}}(x) = \max(x, 0)$$

exhibits the so-called "non-negative homogeneity" property:

$$\forall (z, \alpha) \in \mathbb{R} \times \mathbb{R}^+, \phi_{\text{rect}}(z\alpha) = \alpha\phi_{\text{rect}}(z)$$

The action of a $L$-layered deep feed-forward neural network can be written as:

$$y = \phi_{\text{rect}}\left(\phi_{\text{rect}}\left(\ldots \phi_{\text{rect}}\left(x \cdot W_1 + b_1\right)\ldots\right) \cdot W_{L-1} + b_{L-1}\right) \cdot W_L + b_L$$

in which

- $x$ is the input vector
- $W_L$ is the weight matrix of the $L$-th layer
- $b_L$ is the bias vector of the $L$-th layer

To simplify notation, we have not included the final activation function, which may take any form (softmax or sigmoid etc.) without modification of the proceeding arguments. Generalizing the original arguments from Dinh et al. (2017) slightly to include bias terms, we exploit the non-negative homogeneity of the ReLU function to find that a so-called "$\alpha$-scaling" of one of the layers will not change its behaviour. Explicitly applying this to the $i$-th layer yields:

$$\left(\phi_{\text{rect}}\left(x \cdot \alpha W_i + \alpha b_i\right)\right) \cdot \frac{1}{\alpha}W_{i+1} = \left(\phi_{\text{rect}}\left(x \cdot W_i + b_i\right)\right) \cdot W_{i+1} \qquad (9)$$

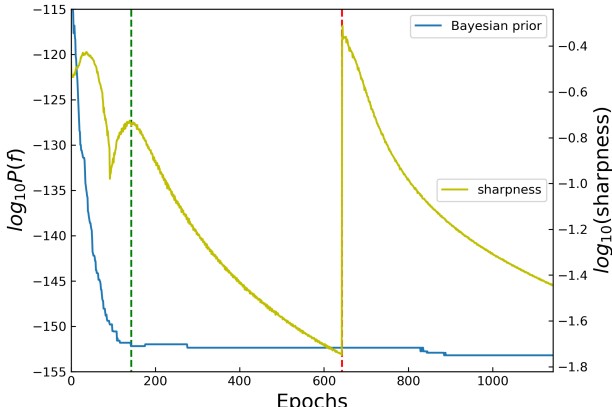

Figure S11: **The effect of alpha scaling on prior and sharpness.** At each epoch we calculate the sharpness and the prior for our FCN on MNIST system with $|S| = 500$. The green dashed line denotes where zero-training error is reached and post-training starts. The red dashed line denotes the epoch where $\alpha$-scaling takes place with $\alpha = 5.0$. Here the value of $\alpha$ is not big enough to "knock" the optimizer out of the neutral space, upon alpha scaling, in contrast to fig. 5. As expected, we observe no change in prior upon alpha scaling (note that prior can change on overtraining if a slightly different function is found by SGD). The sharpness shows a larger peak upon alpha-scaling, as expected. See appendix H.

Clearly, the transformation described by $(W_i, b_i, W_{i+1}) \rightarrow \left(\alpha W_i, \alpha b_i, \frac{1}{\alpha} W_{i+1}\right)$ will lead to an observationally equivalent network (that is, a network whose output is identical for any given input, even if the weight and bias terms differ).

Since the $\alpha$ scaling transformation does not change the function, it does not change the prior of the function. However, for large enough $\alpha$, as shown for example in fig. 5, we see that SGD can be "knocked" out of the current neutral space because of the large gradients that are induced by the $\alpha$ scaling. This typically leads to the prior suddenly surging up, because the random nature of the perturbation means that the system is more likely to land on large volume functions. However, we always observe that the prior then drops back down quite quickly as SGD reaches zero training error again. On the other hand, as shown in fig. S11, when the value of $\alpha$ is smaller it does not knock SGD out of the neutral space, and so the prior does not change at all. Nevertheless, the sharpness still exhibits a strong spike due to the the alpha scaling.

Although not in the scope of this work, it is worth noting that the alpha scaling process in Convolutional Neural Networks (CNNs) with batch normalization (Ioffe & Szegedy, 2015) layer(s) is somewhat different. Because a batch normalization layer will eliminate all affine transformations applied on its inputs, one can arbitrarily alpha scale the layers before a batch normalization layer without needing to of compensate in following layer, provided the scaling is linear.

## G   FLATNESS AND PRIOR CORRELATION

In the main text, we showed the correlation of the Bayesian prior and of sharpness with generalization in fig. 3 and fig. 4. Here, in fig. S12, we show the direct correlation of the prior and sharpness. As expected from the figures in the main text, sharpness correlates with prior roughly as it does with generalization - i.e. reasonably for vanilla SGD but badly for entropy-SGD (Chaudhari et al., 2019) or Adam (Kingma & Ba, 2014). We note that, as shown in fig. S9, sharpness also correlates relatively well with the spectral norm of the Hessian and log product of its 50 largest eigenvalues for all the optimizers. So the correlation of flatness with prior/generalization does not depend much on which particular flatness measure is used.

Overall, it is perhaps unsurprising that a local measure such as flatness varies in how well it approximates the global prior. What is unexpected (at least to us) is that Adam and Entropy-SGD break the correlation for this data set. In appendix J.2, we show that this correlation also breaks down for other more complex optimizers, but, interestingly, not for full-batch SGD. Further empirical and theoretical work is needed to understand this phenomenon. For example, is the optimizer dependence of the correlation between flatness and prior a general property of the optimizer, or is it specific to certain architectures and datasets? One hint that these results may have complex dependencies on architecture and dataset comes from our observation that for ResNet50 on

Cifar10, we see less difference between SGD and Adam than we see for the FCN on MNIST. More work is needed here.

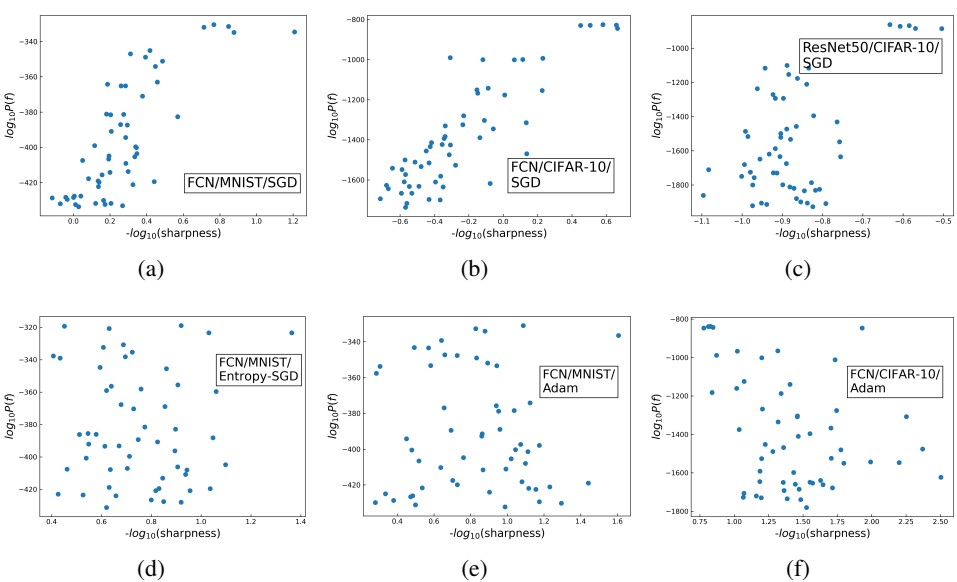

Figure S12: The direct correlation between prior $P(f)$ and sharpness over different datasets and optimizers. The correlation between prior and sharpness closely resembles the correlation between sharpness and generalization, mainly because prior and generalization are very closely correlated, as seen in our experiments (fig. 3, fig. 4).

# H   TEMPORAL BEHAVIOR OF SHARPNESS

When using sharpness in definition 2.1 as the metric of flatness, there are several caveats. First is the hyperparameters (see table 1): the value of sharpness is only meaningful under specified hyperparameters, and in different experiments the sharpnesses are only comparable when the hyperparameters are the same. This renders sharpness less convenient to use (but still much more efficient than Hessian calculation). Second is the time evolving behavior of sharpness: For the classification problems we study, and for CE loss, it can continue to change even when the function (and hence generalization) is unchanged.

Before reaching zero training error, gradients can be large, and the behavior of sharpness (definition 2.1) can be unstable under changes of box size $\zeta$. This effect is likely the cause of some unusual fluctuations in the sharpness that can be observed in fig. 5 and fig. S11 around epoch 100. In fig. S13 we show that this artefact disappears for larger $\zeta$. Similarly, when the gradients are big (typical in training), sharpness may no longer link to spectral norm of Hessian very well.

In fig. S14, we first train the FCN to zero error, then "alpha scale" after 500 epochs, and then keep post-training for another 5000 epochs, much longer than in fig. 5. The behaviour of sharpness and prior upon "alpha scaling" (not surprisingly) follows our discussion in section 5.4. What is interesting to see here is that after enough overtraining, the effect of the alpha scaling spike appears to disappear, and the overall curve looks like a continuation of the curve prior to alpha scaling. What this suggests is that alpha-scaling brings the system to an area of parameter space that is somehow "unnatural". Again, this is a topic that deserves further investigation in the future.

Finally, we show the temporal behavior of a Hessian-based flatness measure in fig. S15. Because of the large memory cost when calculating the Hessian, we use a smaller FCN on MNIST, with the first hidden layer having 10 units. We find that the Hessian based flatness exhibit similar temporal behavior to sharpness.

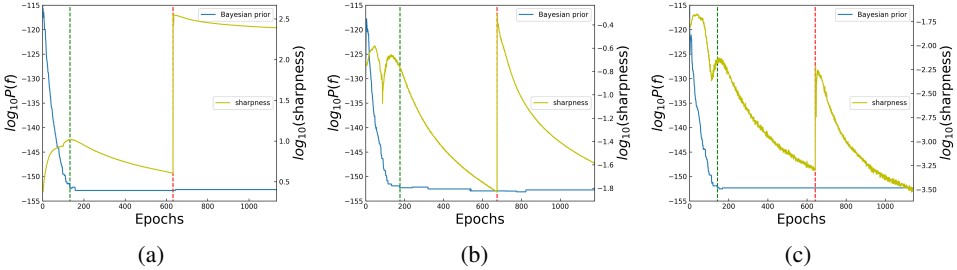

(a)  (b)  (c)

Figure S13:  Different temporal behavior of sharpness, prior and accuracy when using different box size $\zeta$. The dataset is MNIST with $|S| = 500$ and $|E| = 100$. The architecture is FCN. SGD optimizer is used. Scaling parameter $\alpha = 5.0$. Green and red dashed line denote reaching zero training error and alpha scaling, respectively. (a) $\zeta = 10^{-3}$, (b) $\zeta = 10^{-4}$, (c) $\zeta = 10^{-5}$. While there are quantitative differences between the values of $\zeta$ used, qualitatively we observe similar behaviour.

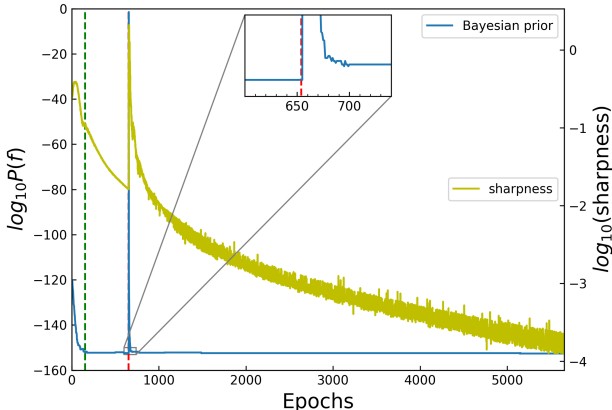

Figure S14: The temporal behavior of sharpness and prior after 5000 epochs of reaching zero training error. The dataset is MNIST with $|S| = 500$ and $|E| = 100$. The architecture is FCN. SGD optimizer is used. The magnitude of scaling $\alpha = 6.0$.

## I  THE CORRELATION BETWEEN GENERALIZATION, PRIOR, AND SHARPNESS UPON OVERTRAINING

As shown in  fig. 5 of the main text, and further discussed in appendix H, flatness measures keep decreasing upon overtraining even when the function itself does not change. In this section, we revisit the correlation between prior, flatness and generalization at different numbers of overtraining epochs, i.e. *after* reaching zero training error.As can be seen in fig. S16 to fig. S21, overtraining does not meaningfully affect the correlation between sharpness, prior, and generalization we observed at the epoch where zero error is first reached in fig. 3 and fig. 4. When the optimizer is SGD, the flatness, no matter if it is measured by sharpness or Hessian based metrics, correlates well with prior and (hence) generalization across difference overtraining epochs; whereas when using Adam, the poor correlation also persist in overtraining.

## J  FURTHER EXPERIMENTS

### J.1  RESNET50 TRAINED WITH ADAM

When training ResNet50 on CIFAR-10, we use training set size $|S| = 5000$, attack set size $|A| = 5000$, test set size $|E| = 2000$. In each experiment, we mix the whole training set with different size of subset of attack set. The size of $|A|$ ranges as $(0, 500, 1000, 1500, ..., 5000)$. For each subset of attack set we sample 5 times. When training ResNet50 with Adam, we empirically found it is hard to train the neural net to zero training error with attack set size $|A| > 2500$. So we only show the results for those functions found with $|A| \leq 2500$. In

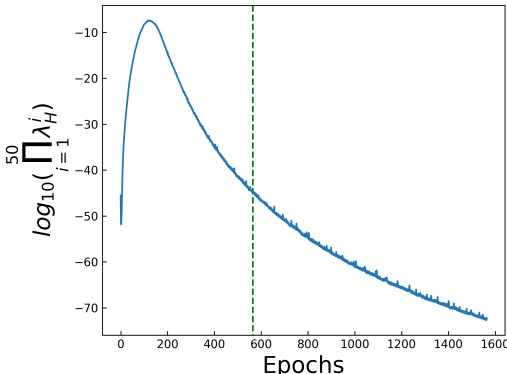

Figure S15: The temporal behavior of one Hessian based flatness metric. The dataset is MNIST with $|S| = 500$ and $|E| = 100$. The architecture is a smaller FCN (784-10-40-1), the optimizer is SGD. The green dashed line denotes the epoch where the system reaches zero training error. No alpha scaling is applied here. The Hessian based flatness metric shows similar temporal behaviour to the sharpness measure.

fig. S22 we show the results of correlation between sharpness and prior with generalization with limited data. The prior, as usual, correlates tightly with generalization, while the flatness-generalization correlation is much more scattered, although it is slightly better than the correlation seen for the FCN on MNIST, and closer to the behaviour we observed for SGD in the main text.

## J.2  MORE SGD-VARIANT OPTIMIZERS

In fig. S23 we provide further empirical results for the impact of choice of optimizer on the sharpness-generalization correlation by studying three common used SGD variants: Adagrad (Duchi et al., 2011), Momentum (Rumelhart et al., 1986) (momentum=0.9) and RMSProp (Tieleman & Hinton, 2012), as well as full batch gradient descent. Interestingly, full batch gradient descent (or simply gradient descent) shows behaviour that is quite similar to vanilla SGD. By contrast, for the other three optimizers, the correlation between sharpness and generalization breaks down, whereas the correlation between prior and generalization remains intact, much as was observed in the main text for Adam and Entropy-SGD. .

## J.3  LARGER TRAINING SET

In order to rule out any potential training size effect on our main argument of the flatness, prior and generalization relationship, we further performed the experiments on MNIST with 10k training examples. Larger training sets are hard because of the GP-EP calculation of the prior scales badly with size. The results are shown in fig. S24. It is clear that the correlations between sharpness, prior and generalization follow the same pattern as we see in fig. 3, in which there are only $|S| = 500, |E| = 1000$ images. If anything, the correlation with prior is tighter.

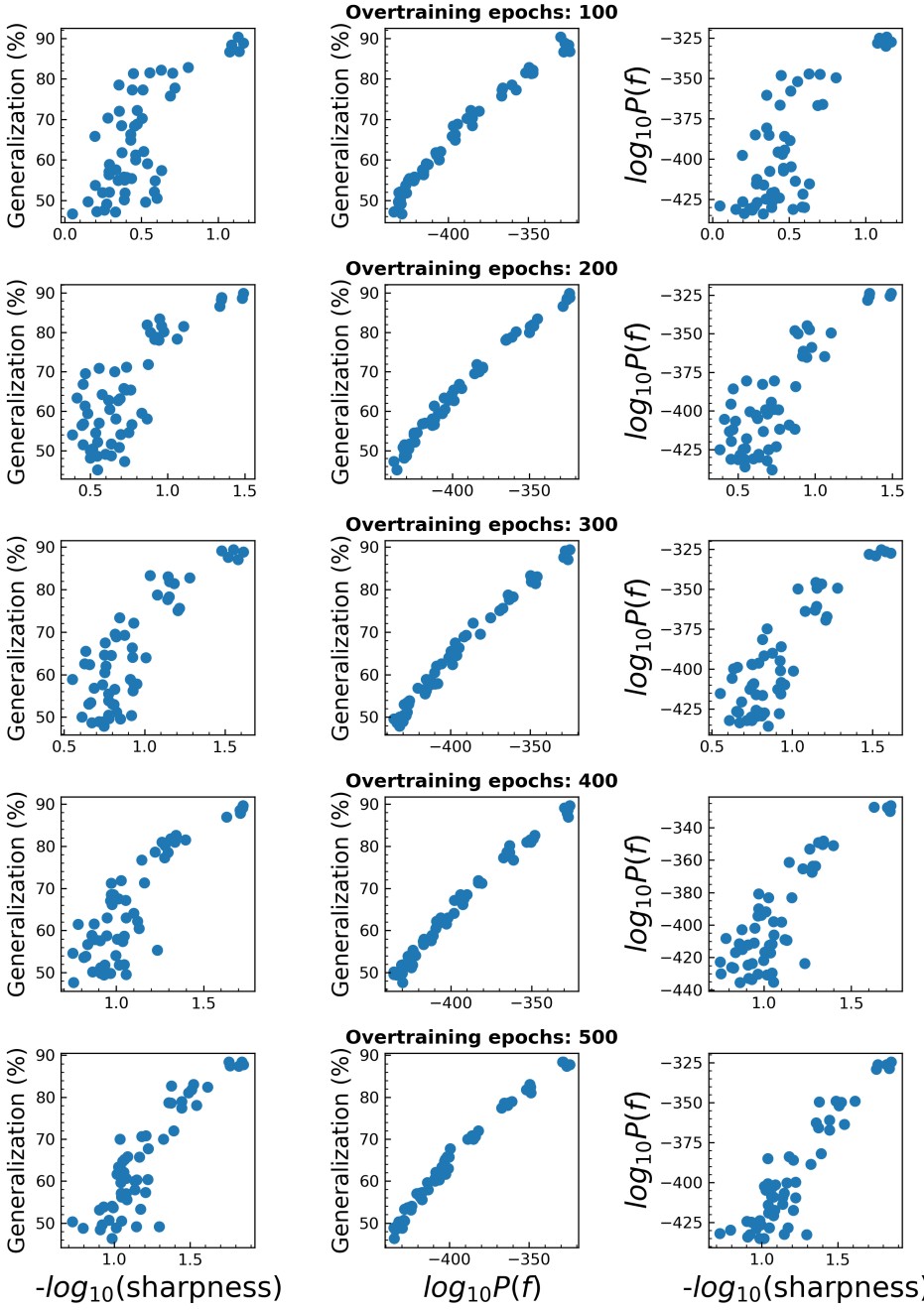

Figure S16: The correlation between sharpness, prior and generalization upon overtraining. The dataset is MNIST ($|S| = 500, |E| = 1000$), the optimizer is SGD. For the range of (100-500) overtraining epoch tested here, the overall values of sharpness drop with overtraining. By contrast, the priors remain largely the same. For each quantity, the correlations remain remarkably similar with overtraining.

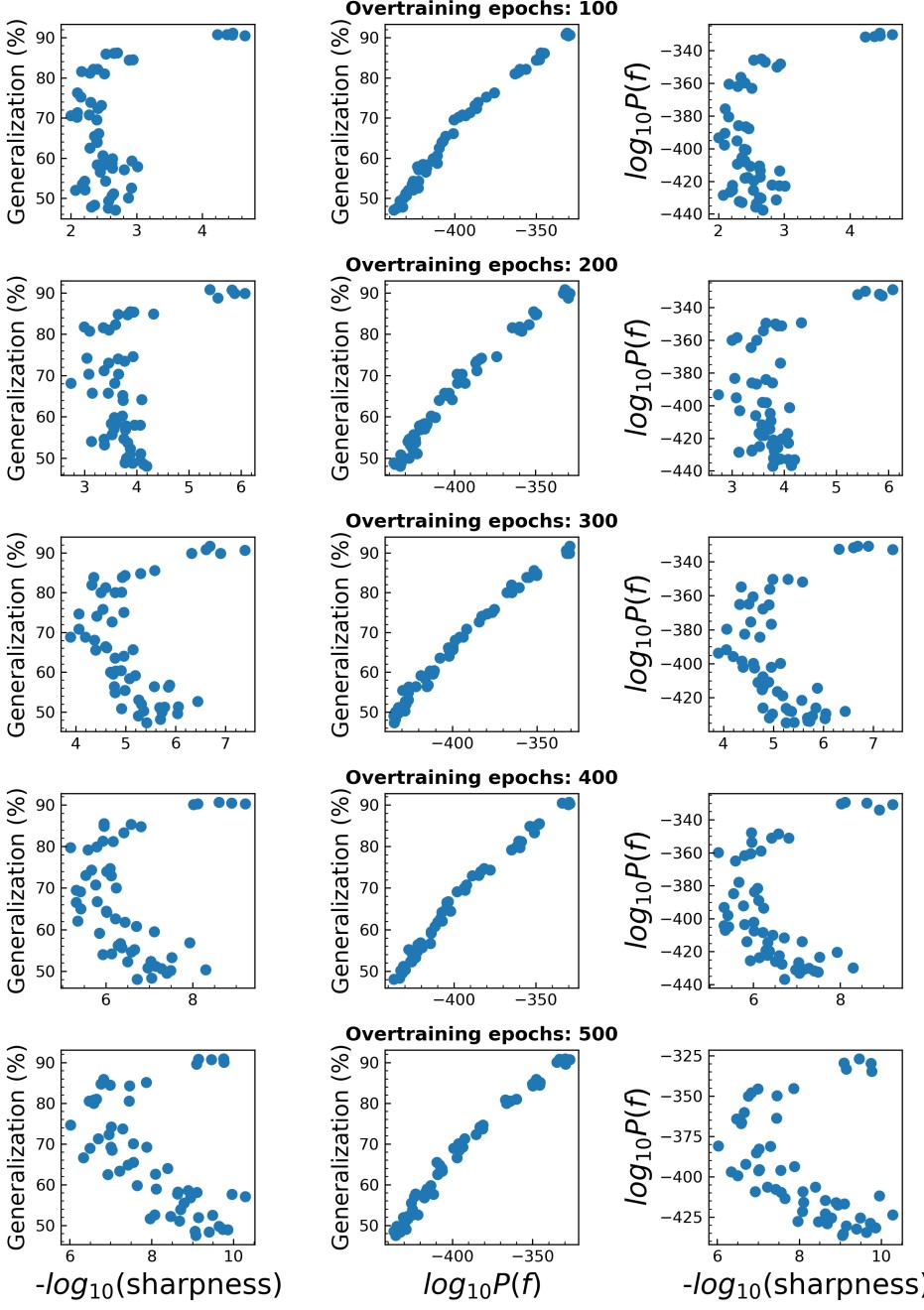

Figure S17: The correlation between sharpness, prior and generalization when over-trained (keep training after reaching zero training error). The dataset is MNIST ($|S| = 500, |E| = 1000$), the optimizer is Adam. The correlations are similar across different overtraining epochs.

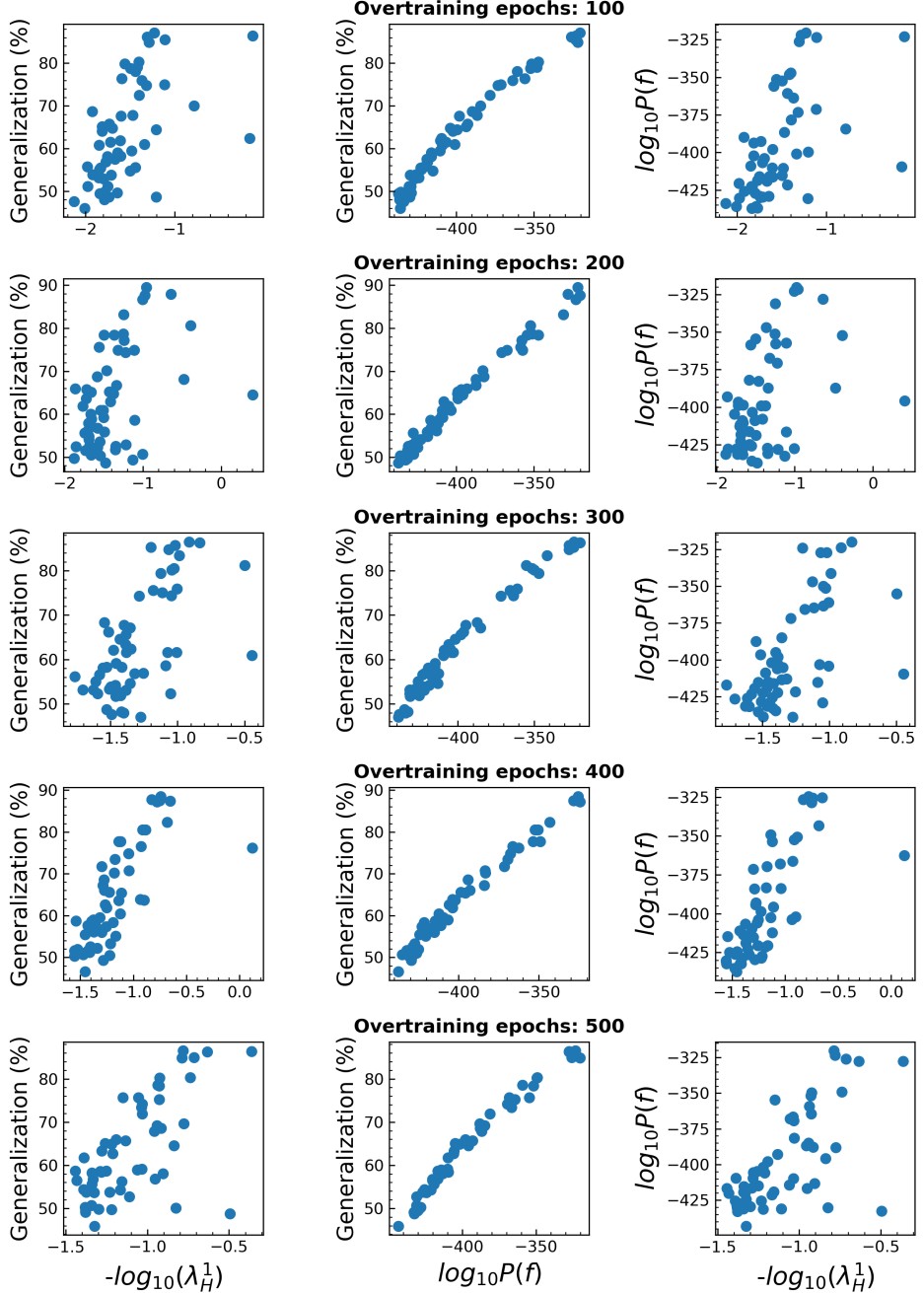

Figure S18: The correlation between Hessian spectral norm, prior and generalization when over-trained (keep training after reaching zero training error). The dataset is MNIST ($|S| = 500, |E| = 1000$), the optimizer is SGD. The correlations are similar across different overtraining epochs.

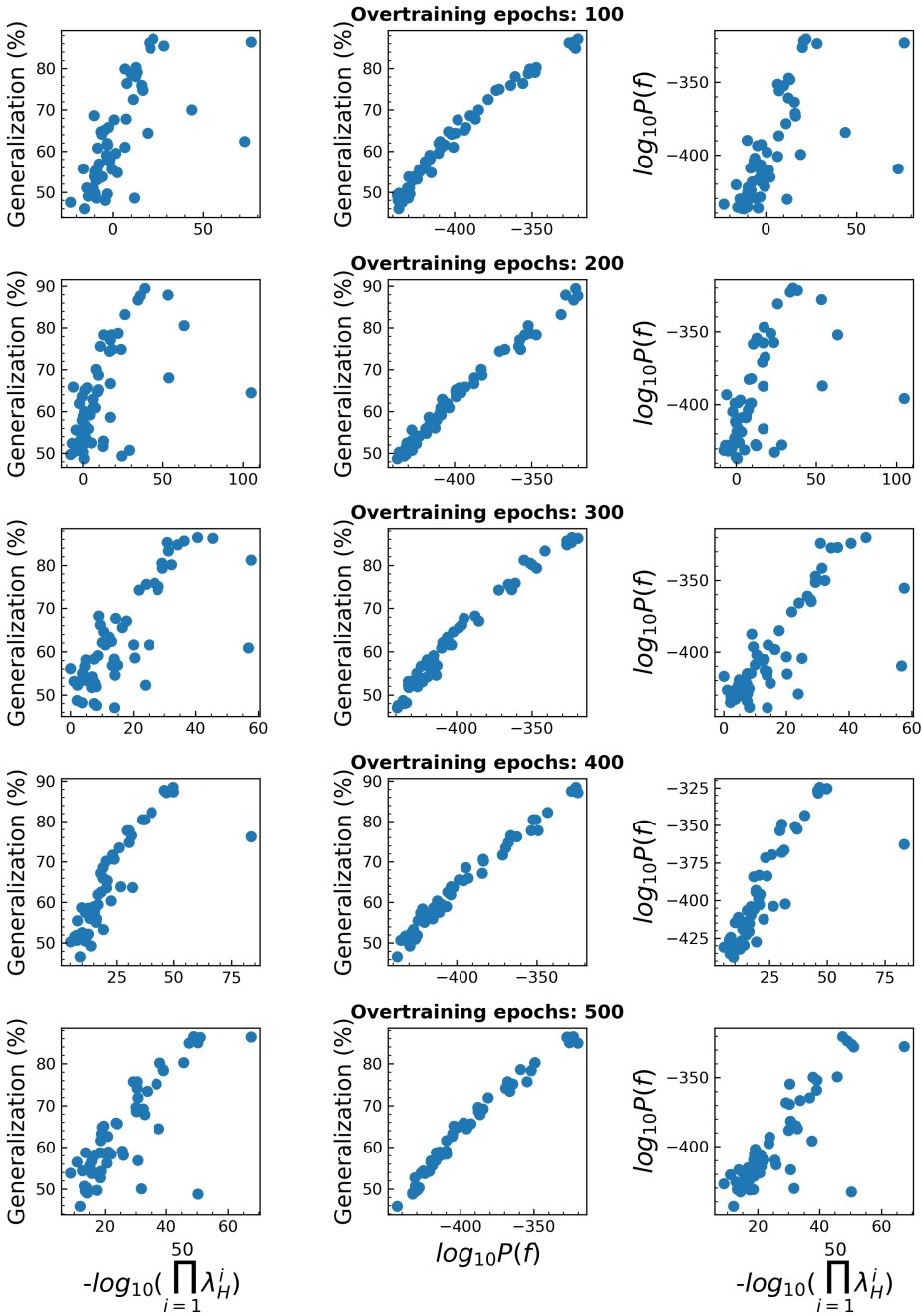

Figure S19: The correlation between Hessian based flatness (product of the top 50 largest Hessian eigenvalues), prior and generalization when over-trained (keep training after reaching zero training error). The dataset is MNIST ($|S| = 500, |E| = 1000$), the optimizer is SGD. The correlations are similar across different overtraining epochs.

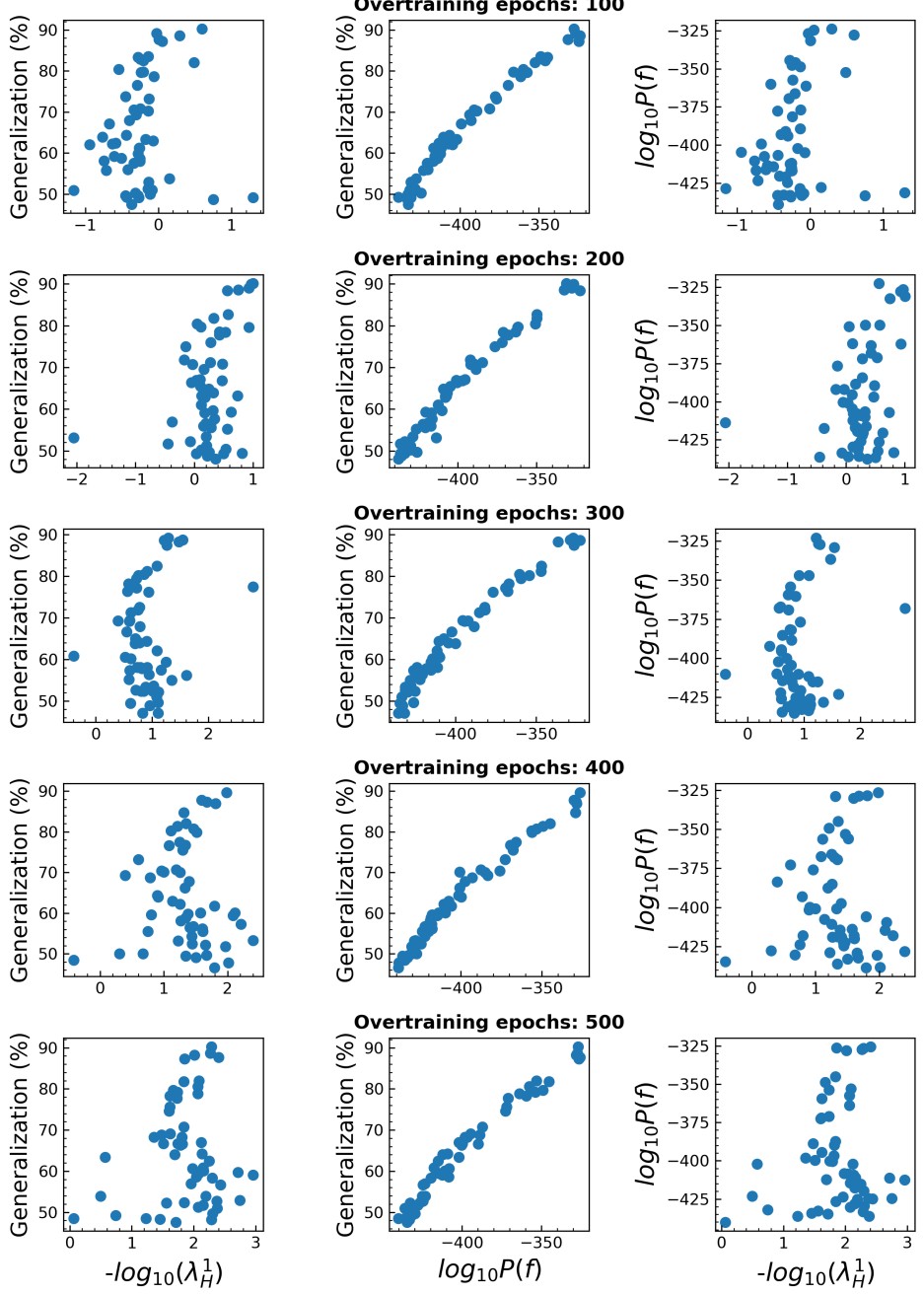

Figure S20: The correlation between Hessian spectral norm, prior and generalization when overtrained (keep training after reaching zero training error). The dataset is MNIST ($|S| = 500, |E| = 1000$), the optimizer is Adam. The correlations are similar across different overtraining epochs.

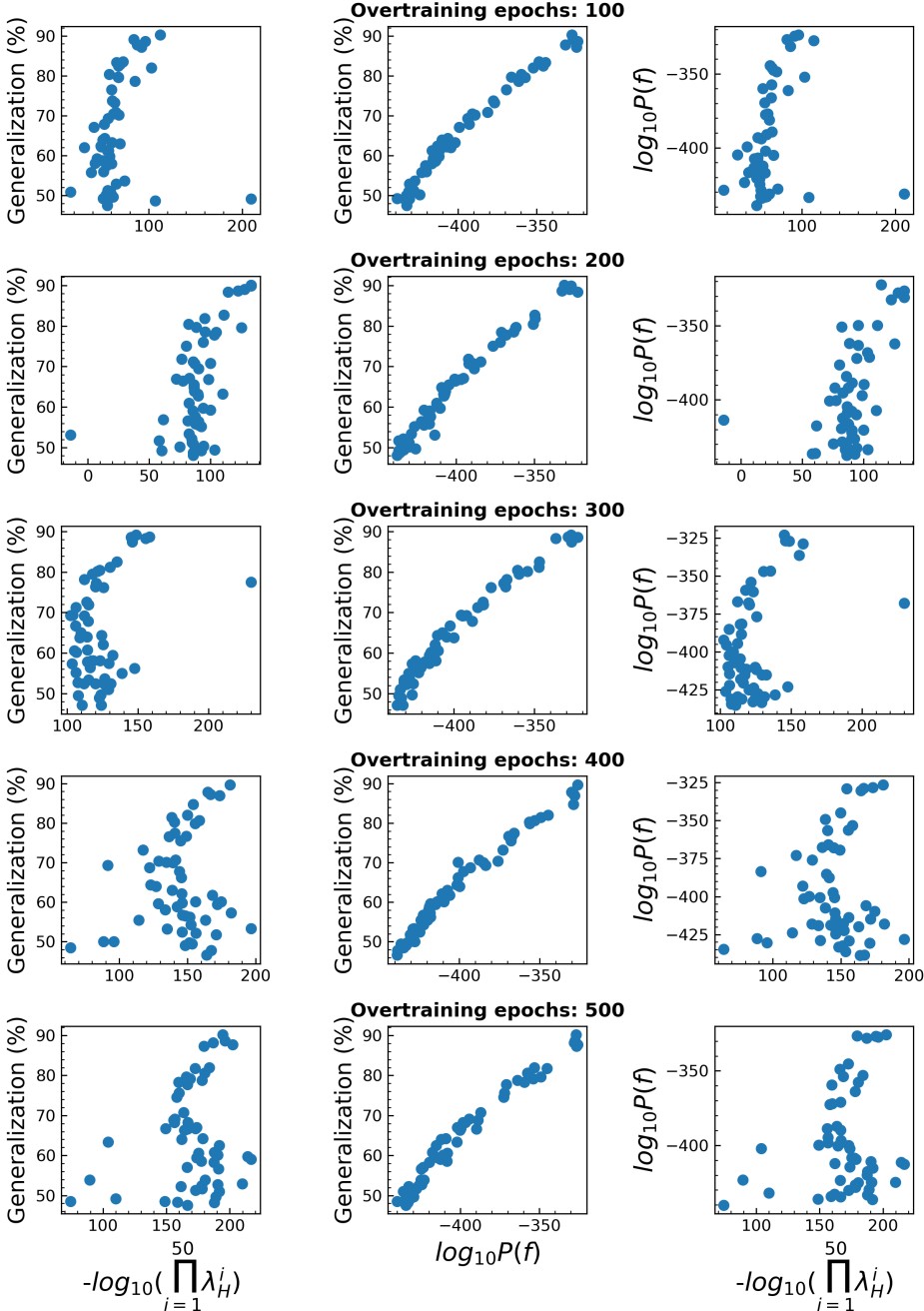

Figure S21: The correlation between Hessian based flatness (product of the top 50 largest Hessian eigenvalues), prior and generalization when over-trained (keep training after reaching zero training error). The dataset is MNIST ($|S| = 500, |E| = 1000$), the optimizer is Adam. The correlations are similar across different overtraining epochs.

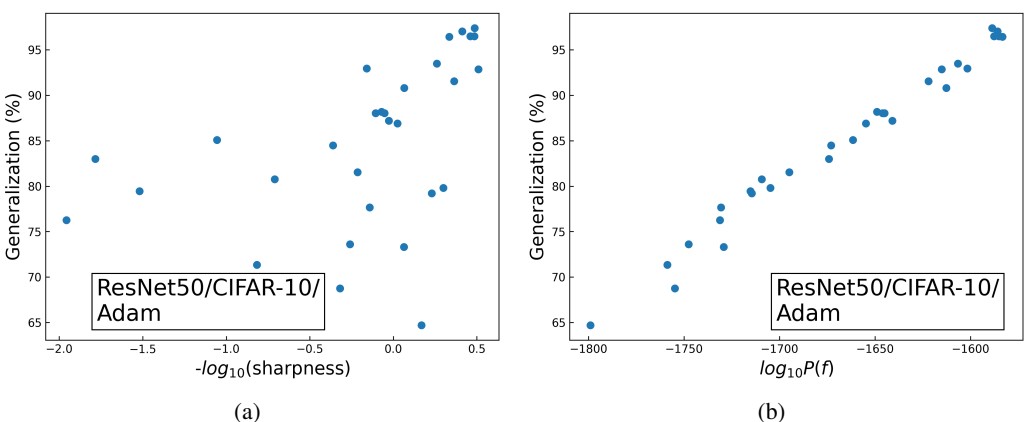

Figure S22: The correlation between generalization and (a) sharpness (b) prior for ResNet50 with $|S| = 5000$, $|E| = 2000$, and $|A|$ ranging from 0 to 2500, all on CIFAR-10.

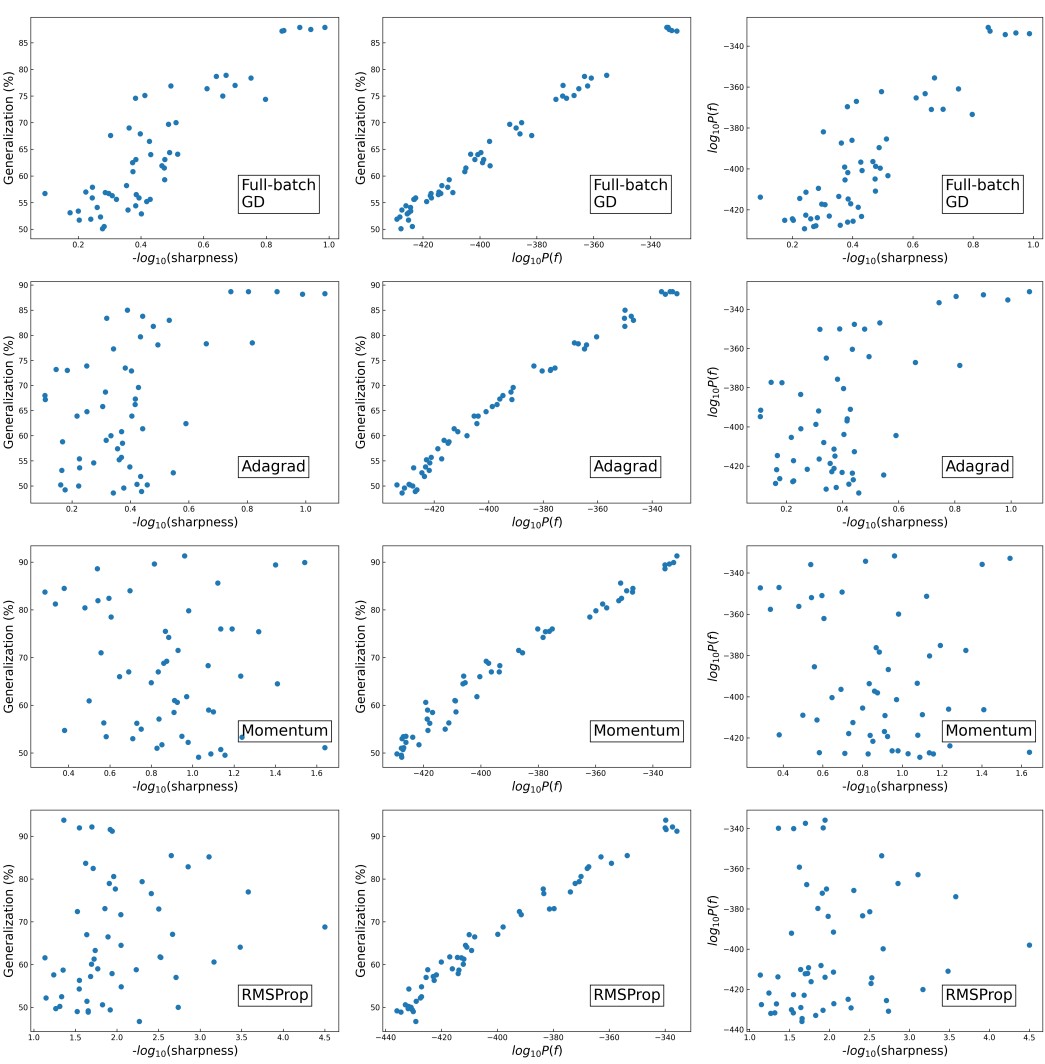

Figure S23: More results on the correlation between sharpness, prior and generalization when using other SGD-variant optimizers. The dataset is MNIST, $|S| = 500, |E| = 1000$. The architecture is FCN. The optimizers are full-batch gradient descent, Adagrad, Momentum (momentum=0.9) and RMSProp. All correlations are measured upon reaching zero training error.

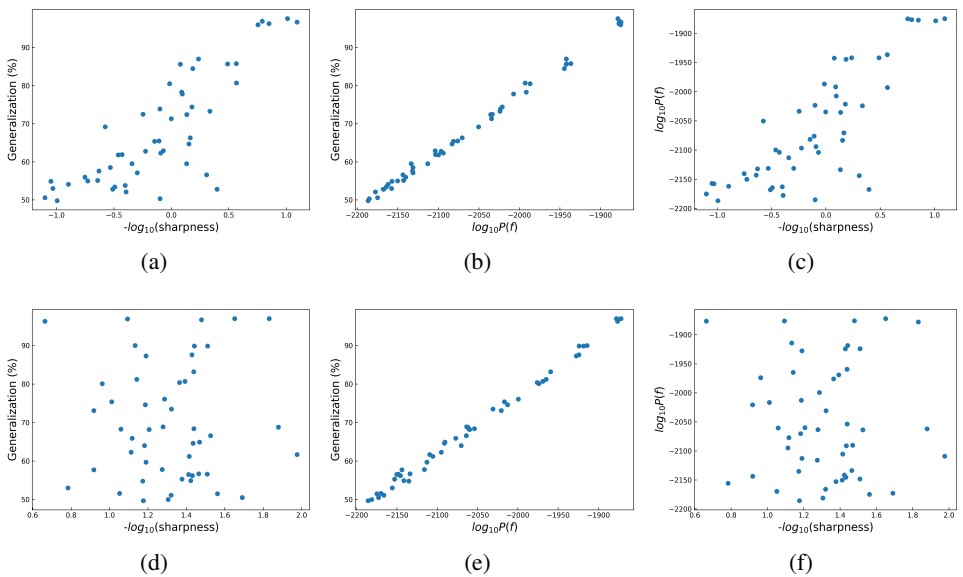

Figure S24: The correlation between sharpness, prior and generalization on MNIST with $|S| = 10000, |E| = 1000$. The attack set size ranges from 1000 to 9000. The architecture is FCN. (a)-(c): The FCN is trained with SGD; (d)-(f): The FCN is trained with Adam.

