# OpenReview forum: "WHY FLATNESS DOES AND DOES NOT CORRELATE WITH GENERALIZATION FOR DEEP NEURAL NETWORKS "
_ICLR.cc/2022/Conference — ICLR 2022 Submitted_

### Official Review · Reviewer_hYg1 · 2021-10-20

**Correctness:** 2
**Technical Novelty And Significance:** 2
**Empirical Novelty And Significance:** 2
**Recommendation:** 3
**Confidence:** 4

**Main Review:**

First, its extremely easy to determine the authors. They self-cite collectively over 50 times if you include the publications where the authors are co-authored. Furthermore, they have the current and the former NeurIPS version of this paper posted on arxiv, where the original title was slightly different and with a less ambiguous conclusion -- "Why Flatness Correlates With Generalization For Deep Neural Networks". This isn't against ICLRs rules, and this review is not to judge either arxiv version, but since v1 is publicly available, I have considered it among the relevant past work for the currently submitted paper.

General Concerns (no specific order):

	- The name "neutral space of f" has a much more commonly known name among the mathematics/neural network community, its called "the fiber of f". See https://en.wikipedia.org/wiki/Fiber_(mathematics).

	- Section 2.4, it is stated that "DNNs are typically trained to zero training error on the training set". As a practitioner myself, I'm not sure who/where this information is coming from. DNNs are typically *not* trained to zero training error as that almost always results in overfitting and an over-usage of time/computational resources.

	- Figure 1, it shows the loss function has negative error at certain weights (w). This seems to contradict what was said about loss functions in the paragraph immediately prior, that the loss function is zero or positive. What does negative error mean?

	- I find it slightly disingenuous to place central definitions in the appendices to achieve the page limit. Gamifying the page limit isn't the purpose of the page limit. Expecting reviewers to read 35 pages in the time allotted isn't considerate.

        - Appendix B where the "parameter-function" map is defined, it states that it "was first introduced in [Valle-Pérez et al., 2018]". The map from the parameter space which takes a parameter and maps it to a function is so ubiquitous its questionable to assign any single person as its creator. My cynical side questions if this was an attempt at self-citation.

	- Minor: The text makes the jump from L(y,y') to L(w) without stating that y is parameterized by w. Its not a big deal, but the less familiar reader may get confused at an early stage with this jump.

	- Section 3 states "Perhaps the simplest argument is that if DNNs trained to zero error are known to generalize well on unseen data...". That's a very big "if". In practice, DNNs trained to zero error are expected to not generalize well on unseen data.

	- Section 3 states "Mingard et al. (2021) showed that log (P_B(f|S)) correlates quite tightly with generalization error." If this is the case, then its not clear to me if the second aspect of this work is a novel contribution. If it is, its not clear to me how.

	- Section 5.1 states that the DNN being used "was found to be sufficiently expressive to represent almost all possible functions". The ReLU network is shaped 7 - 40 - 40 - 2, and its claimed that this is capable of representing "almost all of the possible" 3.4 x 10^34 functions. I'd like proof. For example, consider proposition 3 from the well known work on ReLU linear regions https://arxiv.org/abs/1402.1869, where even the most conservative upper bound for the number of functions computable by a ReLU network with N hidden neurons is 2^N = 2^80 = O(10^24). This is 10 orders of magnitude short for this case. The lower bound on the maximum (Theorem 4) is even more pessimistic, and both of these bounds have been shown to be overoptimistic in later works (e.g. https://arxiv.org/abs/1711.02114) taking advantage of the fact that zero-activations of ReLUs collapse the dimension of the signal passing through the network. Ten orders of magnitude is hard to imagine.

	- Section 5.1, the authors take 10^8 initial points. That is 26 orders of magnitude short of how many functions are computable, or 0.000000000000000000000001% of the possible functions would've been sampled. It is likely that even less were sampled since these points were chosen from essentially a bounded sphere in parameter space (i.e. finite samples from a thin-tailed Gaussian distribution). Its difficult to back up any empirical claim when there's 10^34 functions.

	- Critical: Section 5.2 it is stated that "In order to generate functions f with zero error on the training set S, but with diverse generalization performance, we use the attack-set trick from Wu et al. (2017)." This seems like a critical flaw in that the training process is not SGD, and therefore the distribution of parameters could be substantially different than that of using SGD. This feels like the evidence only supports the different claim that "for this attack dataset trick, these claims about flatness (volume) measures hold".

	- Section 5.3 states "Changing optimizers or changing hyperparameters can, of course, alter the generalization performance by small amounts, which may be critically important in practical applications". True and False, if "small" was replaced by "large", I'd say true. To be facetious, one could choose hyperparameters extreme enough where generalization is arbitrarily poor. Since hyperparameters can alter the level of generalization from arbitrarily poor to state-of-the-art for some models, this is an important point. Perhaps its present somewhere in the 35 pages and I do not see it, but I'm unable to find much comparison for this at all.

	- Figure 5, what is "alpha scaling"? Not introduced as far as I can find.

	- In the discussion, it states "evidential basis for this hypothesis has always been mixed". It seems like the first main contribution of this work adds to the pile of mixed results. Another excerpt, "Here we performed extensive empirical work showing that flatness can indeed correlate with generalization. However, this correlation is not always tight, and can be easily broken by changing the optimizer, or by parameter-rescaling". What I take away from this is that sometimes its correlated, sometimes its not, and there's no clear pattern at this point. Seeing as this is the main title of this paper, it doesn't feel like a novel result at all. It seems like more supporting evidence that we really don't know how flatness measures correlate with generalization. I could be wrong, clarification would be helpful.

Typsetting:

	- Many figures are hard to read, with ticks have <6 pt font, or otherwise appear to have errors in typesetting, graphs sitting overtop of words, or maybe labels were manipulated on MS paint or the alike. Overall there's very little consistency across figures.

	- Figure size varies substantially, legends are seemingly randomly placed, sometimes larger than the labels sometimes not. Figures are different sizes even within the same 8-subfigure figure.

	- SOTA, state of the art (or also written state-of-the-art in this same paper), doesn't require an abbreviation since its used once. Well really, its introduced twice, along with its abbreviation twice. Regardless, not needed. Same goes for cross entropy (CE), introduced without ever using it again. Gaussian Processes (GPs) is used multiple times, but also introduced multiple times, and its not introduced at the first opportunity either. Seems unorganized.

	- The use of w.r.t., Just write it out instead of bringing in needless abbreviations.

	- Quotation style fluctuates often. Look closely at the frequently changing quotation marks.

	- fig. vs. Fig., Appendix vs. appendix, Eq.(4). vs. Eq. (4). Figs vs. fig. for multiple subplots. Consistency and generally speaking, upper case "Fig." and "Eq." tend to look more professional.

	- General equation typesetting, spacing, periods before a sentence with an equation ends.

	- Commas, spaces, just simple typos that could be caught by any spell check.

	- fig. 2 vs. fig. S3, I don't get the S's. For example, in Section 5.1, it states the figure is in the appendix but its linked to regular Fig. 3. Sometimes figures with "S#" link to the main text, sometimes not. Check this.

	- "The function "with" the largest prior...". Missing the word "with".


**Summary Of The Paper:**

This paper shows that for certain variants of SGD, the commonly held assertion that flat minima imply good generalization may not hold. The overall conclusion from this aspect of the paper is that popular measures of flatness sometimes do, and sometimes do not, correlate with generalization. The authors continue on to propose an alternative measure, the Bayesian prior, which they claim correlates well with generalization. In essence, this measure captures the volume of parameter space which when a networks weights are initialized within the volume's region, it converge to a pre-specified function with low training error. After covering the definitions and related measures, they perform experiments on MNIST, CIFAR-10, and discuss how the choice of optimizer affects the claims.

**Summary Of The Review:**

Overall, its very long, there are a few critical concerns about the soundness of the claims, there are some typesetting issues. The figures could use work, a wall of dozens of hard-to-read graphs at the end aren't helping the focus. The conclusion for me is quite ambiguous, I'm not sure what to take away from this paper in terms of if flatness makes sense to use as a predictor of generalization. The title doesn't entirely reflect what's being claimed in the paper, since much of the paper focuses on the Bayesian prior.

---

### Official Review · Reviewer_r1hF · 2021-11-02

**Correctness:** 4
**Technical Novelty And Significance:** 1
**Empirical Novelty And Significance:** 2
**Recommendation:** 3
**Confidence:** 4

**Main Review:**

Strengths:
- The empirical observation that Adam and Entropy-SGD could break the correlation between flatness and generalization is interesting.
- The empirical observation that the Bayesian prior is linked with generalization, reinforces the ideas presented in the Paper (Mingard et. al., 2021). This is an interesting area to explore, especially to connect empirical observations with theoretical findings.
- Throughout the paper, there are some nice explanations/intuitions given.
- For each figure presented in the main paper, there are similar experiments to provide sanity checks.

Weaknesses:
- The novelty of the paper is very limited:
    - Previous work already suggests that care must be taken when interpreting local measures of flatness. In [1] it has been shown that local flatness alone is not enough, and we need global well-connectivity as well.
    -The results regarding how re-scaling would break the correlation between generalization and flatness are also studied before in the literature (Dinh et al., 2017).
    -The link between the Bayesian prior and generalization is studied in (Mingard et. al., 2021). More precisely, in (Mingard et. al., 2021), the authors show that $P_B (f | S)∝$ generalization error. As for a given $S, P_B (f | S)∝ P (f)$, then it is implicit that $P (f) ∝ $ generalization error, which is the main result of this submission.
- The paper needs to be fully proofread. There are many typos throughout the paper (missing "a", "the", MINST instead of MNIST, the first sentence of Section 5.3 is incomplete, etc.). The flow of the paper also needs major adjustments. The paragraphs are not coherently written. Also, a lot of the sentences could be summarized/re-written. Some example suggestions:
   - Definition 2.2 is not needed.
   - In contribution number 2, explanations should be avoided. Instead only the main results should be stated. From the sentence "For discrete .... " the text should be moved to the proper section.
   - In some parts of the text, some unusual terms are used. Examples: Section 2.1, "the inputs live in ", "the observed output y".
   - The parameter vector w is not defined in Section 2.1, and in Section 2.2 the loss function is a function of w; which is a mismatch with the notation used in Section 2.1 to define the loss function.
   - Section 4 could be combined with Section 3.
   - There are repetitions in Section 3 and some paragraphs do not follow well from one another.
   - As there are no theoretical results, Section 2 could be significantly shortened.

   All these would help with the flow of the ideas and would help the reader to stay focused on the main outcomes of the paper. In the current shape, the main outcomes are in the shadows: the results of the paper only come after page 6.

- As the paper is an empirical study, it is lacking important ablation studies. The scope of the empirical results should be expanded (for example be similar to the scope of empirical results in [1]). One example case study is to compare different initial random seeds: are the networks with higher volume (higher Bayesian prior) also the networks with higher test accuracy?

- It would be better to move the last paragraph of the related work (which is currently in the appendix) to the main paper, and discuss exactly the new contributions in light of the three mentioned prior work. It is implicit that if the Bayesian posterior is related to generalization (which is studied thoroughly in (Mingard et al., 2021)), then the Bayesian prior is also related to generalization. Stating this result is not a new claim. Although I believe that empirically showing this is interesting, for this paper to be an empirical study, many more ablation studies and settings should be studied to make the claim practically convincing.


Minor comments:
- There is a (rather new) study on neural network loss landscapes [1], where the authors show that the best accuracy is achieved when not only the model has converged to a locally flat region but also to a globally well-connected region. It would be interesting to compare the Bayesian prior with the combination of global connectivity and local flatness. What is the difference between global connectivity and Bayesian prior? Is there a relation between the two? Can we say that globally well-connected regions also have a large "volume"?
- Similar to Definitions 2.1 and 2.2 which are properly cited, definition 2.3 should also be cited from (Mingard et al., 2021).
- Figure 1 is a replica of figure 6.a of (Mingard et al., 2021). Proper credit should be given.
- Throughout the paper, either only refer to flatness, or only refer to sharpness. The inconsistency makes the results/text less readable.
- The results in Figure 2 are not the main focus of the paper. Instead, the main focus is to find scenarios where the correlation between the two and therefore the correlation between flatness and generalization breaks. So, there is no need to present Figure 2 in the main paper.

[1] Yang et al., Taxonomizing local versus global structure in neural network loss landscapes, 2021






**Summary Of The Paper:**

This paper studies the relation between generalization and the flatness of the loss landscapes. The authors show that the correlation between local flatness and generalization is broken in the case of using Adam or Entropy-SGD or by performing parameter rescaling. They show that the Bayesian prior of DNNs (which are trained until zero training error) could on the other hand predict generalization better than the local flatness measures. This is motivated by the work of (Mingard et. al., 2021), where it is shown that the Bayesian posterior function correlates well with the probability function of DNNs that reach zero training error using SGD.

**Summary Of The Review:**

Overall, although the two observations that the Bayesian prior is empirically linked with generalization and that when using Adam or Entropy-SGD flatness is no longer linked to generalization are interesting, the contribution of the paper is not enough. The novelty of the work is very limited and the empirical results are insufficient.

---

### Official Review · Reviewer_z9TP · 2021-11-02

**Correctness:** 3
**Technical Novelty And Significance:** 2
**Empirical Novelty And Significance:** 3
**Recommendation:** 5
**Confidence:** 3

**Main Review:**

There are several recent works, which looked into the connection between generalization and the sharpness of the local minimum. Here, 'sharpness" is often defined using the spectrum of the Hessian (in addition to some normalization terms in some cases). However, previous measures of sharpness are not invariant under parameter rescaling, which would produce the same solution but with a different level of sharpness.

The authors in this work propose an alternative measure: the log of the prior of the function as a predictor of generalization. The motivation behind this comes from recent lines of work which look into the relation between generalization and the posterior probability p(f | S). The authors note that the posterior is related to the prior p(f) by a constant term since neural networks are expressive enough to have zero training error. Hence, p(f) might serve as a good predictor of generalization. Here, the prior on the 0-1 function f is the total probability mass of the weights that produce that function over a fixed dataset (training + test), where the probability measure over the weights is computed using GP. The authors demonstrate experimentally on MNIST and CIFAR10 that there is indeed a strong correlation between p(f) and generalization. In addition, because this is defined over 0-1 functions, it is invariant to parameter rescaling.

In general, the paper is well-written and provide a nice historical review. However, it is not clear where the novelty/contribution is. As mentioned by the authors, previous works have already shown that p(f|S) is a good predictor of generalization. The only addition in this work is replacing p(f|S) with p(f). But, because p(f) is a constant multiple of p(f|S), claiming that p(f) correlates well with generalization is the same as claiming that p(f|S) correlates with generalization, which is what was previously established.

In addition, I do have concerns with the way p(f) is computed. In this paper, the authors use SGD as a way of sampling function f. If there exists good functions that are not often found by SGD, the space of functions studied by the paper would be a distorted version of the reality. For example, some works on reinitialization (e.g. https://arxiv.org/abs/2109.00267) suggest that there exists flatter local minima that are only reached by SGD if a subset of the parameters are reinitialized frequently. In other words, a good local minimum would be surrounded by sharper local minima that generalize less, but because the flatter local minima are surrounded by the sharper ones, SGD would settle on the sharper local minima. This is fixed using reinitialization. If one uses SGD to sample the functions f, many functions could be missing.

Other comments.

Questions:
- How is sharpness in Definition 2.1 computed in the experiments?

Minor issues:
- In Definition 2.1, it should be mentioned that w is a stationary point, otherwise the connection to the Hessian is wrong.
- In Definition 2.2, the actual definition is on a separate paragraph.
- The definition of p(S) can be misleading. It is a function of S, but not "probability" of the sample S. I suggest using a different notation (e.g. g(S) instead of p(S)).
- The statement in Page 4 related to Figure 1 is not really justified. One can draw a picture in which that statement does not hold (i.e. that the volume of f is not correlated with p(f). It is just an intuition but it can be wrong and it is not clear to me why it is more likely to be true. Within the main text, the authors use a more acceptable argument about p(f) having different orders of magnitude but that's different from the claim in Figure 1.
- Please move Figure 2 to the top of the page.
- I am curious to know how SGD with momentum and the recent Sharpness-Aware Minimization (SAM) would perform in Figure 4. Have the authors experimented with them?


Typos:
- Page 6, "The function the largest prior".
- Page 9: "needed.Morever" --> needs a full stop.

**Summary Of The Paper:**

The paper proposes using the log of the prior of the function as a predictor of generalization. They provide arguments for why this is better than using sharpness of the local minima, and support their arguments with empirical results on small datasets.

**Summary Of The Review:**

The paper is well written. However, the main contribution in the paper seems to be identical to previous works. I would appreciate it if the authors elaborate on what their contribution is compared to the works of Valle-Pérez & Louis (2020) and Mingard 2021.

---

### Official Review · Reviewer_ohft · 2021-11-02

**Correctness:** 3
**Technical Novelty And Significance:** 3
**Empirical Novelty And Significance:** 3
**Recommendation:** 5
**Confidence:** 4

**Main Review:**

**Strength**: This paper is relatively well written and easy to follow. It studies an interesting and important question that characterizes the generalization performance of a trained neural network. It proposes a new criterion and demonstrates the effectiveness in standard datasets.

**Weakness**: I think the paper is still a bit weak in a number of aspects:

1. Optimizers plays an important role in deciding the final form of a trained model, but the architecture and parameterization are also fundamentally important. It would be great if in addition to the optimizer comparison experiments, the paper could add experiments to compare *across* different architectures. There is also a belief that overparamerization contribute to better flatness (and generalization). So it would be good to see how the number of model parameters correlate with flatness and the Bayesian prior as well.

2. There is a new optimizer called Sharpness-aware Minimization ( https://arxiv.org/abs/2010.01412 ) that minimize the loss sharpness during optimization and are shown to greatly improve the generalization performance in some tasks. It would be interesting to add SAM to the existing optimizer comparison experiments.

3. Although not strictly required, developing a theoretical link between the proposed criterion and generalization would make the paper more coherent and solid.

4. GPs are used to approximate the Bayesian posteriors in this paper. How accurate is this approximation? Could the paper provide a synthetic study (like in 5.1) where the groundtruth can be measured by direct sampling and compare the accuracy of GP based estimation?

5. In the correlation figures (e.g. Fig. 3), the generalization is shown as the y axis. I wonder are all the networks trained to 0 training error? If not, what would be the plot like when test accuracy is plotted in the y axis?


------------------
After rebuttal: Thanks for the clarifications. I intend to keep my rating.

**Summary Of The Paper:**

This paper studies the correlation between flatness and generalization in deep neural networks and show that, consists with some previous studies, the correlation could sometimes be broken. As an alternative, it propose a new measure based on the Bayesian prior upon initialization, and empirically demonstrate this criterion could maintain a positive correlation with generalization even in the case where flatness breaks.

**Summary Of The Review:**

This paper propose a new criterion to measure generalization and empirically show that it correlate with generalization, even in cases where flatness fails to maintain the correlation. It would be more solid if direct theoretical link could be shown. It is OK for the paper to be primarily empirical studies, but in this case, a more systematic set of experiments could help strengthen the paper. While it is impossible to enumerate all possible comparison cases, a number of canonical settings like different parameterizations (see above for details) are very important.

---

### Official Review · Reviewer_ZCFx · 2021-11-04

**Correctness:** 3
**Technical Novelty And Significance:** 2
**Empirical Novelty And Significance:** 3
**Recommendation:** 6
**Confidence:** 3

**Main Review:**

## Strengths:

1. In the specific settings considered, the correlation between generalization and the proposed Bayesian prior is extremely strong, and robustly outperforms the flatness measure.

1. Appendix contains many sub-settings considering different measures of flatness and different training durations, all corroborating the findings in the main text.

1. Disentangling the influence of architecture and optimization on generalization is an important area of research.

## Weaknesses:

The main weakness of the paper is that all measurements are done under a somewhat exotic setup, that IMO does not resemble a realistic situation where one would do model selection based on some generalization measure. For this reason I find strong claims about the Bayesian prior being a more robust predictor than flatness (e.g. in the abstract and in the conclusion) unsubstantiated. Precisely:

1. The authors use attack sets to generate models with 0 training loss of varying generalization. I suspect this might produce a very specific distribution of resulting functions. A much more natural and realistic setup would be to just sample many different hyper-parameters like learning rate, batch size, learning rate decay, momentum, number of epochs, weight decay etc. By pushing them to extremes, I am pretty sure it is possible to obtain 100%-train accurate yet poorly-generalizing models. And even if they all happen to generalize well, then that resulting smaller range of performance would be the realistic scale at which practitioners and researchers would want to see a correlation between generalization and the proposed proxy. Another more realistic parameter to vary would be just the training set size, since training on/obtaining more data could be costly, and one might want to estimate generalization scaling as a function of training set size. But an attack set setting is something that I find much less practically relevant, and worry that if we were to consider a practical setting as above, the correlation might not be there (since the manifold of models trained with varying attack sets and the more practically-relevant manifold of models trained with varying hyperparameters are likely very different). Please correct me if I'm wrong, or missing something.

1. IIUC the Bayesian prior should only be compared between the same models due to the assumption that $P(S)$ is constant (which is otherwise intractable). Therefore $P(f)$ cannot be used for model selection between different architectures or priors on the weights. Only the training set and the training procedure can vary. In contrast, flatness can be compared between any NNs with the same loss function, which is another reason why I think the claim about prior being a superior measure is too strong.

1. Finally, I think it's realistic to expect that there are manifolds of models where prior does not correlate with generalization, and it would be nice if the authors proactively considered / discussed such settings, to set more realistic expectations for the reader. For example, I suspect that in a regression / MSE loss setting, you could obtain a model that fits the training set perfectly, but then falls off to 0 rapidly everywhere else and doesn't generalize at all. Yet this model will have the highest prior possible, due to the prior being a zero-centered Gaussian. In a classification setting, perhaps constructing an attack set differently could also lead to different behavior (e.g. suppose I only misclassify one class incorrectly, i.e. return a constant class label for the whole attack set - would the prior still correlate with generalization then?). It's fair to leave this sort of investigation for future work, but without such I again find the central claim of the paper too strong.


## Questions:

1. Is the sharpness measured on the training set, or the training and attack set jointly, and do you think it matters? Have you, or any prior work considered measuring sharpness on a separate validation set, and do you think it would make sharpness perform better?

2. Have you considered similar experiments in an MSE / regression setting, with continuous-valued outputs? In this setting it would be simpler and cheaper to measure $P(f)$ in closed-form with NNGP, and I'm curious how MSE and accuracy would correlate with the prior / flatness in this case.

## Minor:

1. Abstract: "zero error on a test set" -> "training set"?
1. Section 2.1: the loss is defined on the space of outputs $\mathcal{Y}$, but is usually instead noted as $L(\bf{w})$. This is inconsistent with definition but also makes it hard to understand if it's the train / train + attack / validation set loss.
1. Page 5, line 2: $P_{SGD}(f|S)$?
1. Figure 2 caption: "The function _with_ the largest prior"?
1. Figure 2, regarding two bands: could you add different colors for them?
1. Typos in section A.1, e.g. "Wwe",  "type B>".


**Summary Of The Paper:**

In binary image classification setting, the paper shows that when using an attack set (augmenting the training set with additional intentionally mislabeled datapoints) to vary the generalization performance of a fixed neural network (NN) architecture, the (approximation of) Bayesian prior $P(f)$ of the outputs of the trained NN on the test set correlates _very_ well with generalization. The effect persists when using different optimizers, and different number of training epochs.

In contrast, the popular measure of flatness (approximation of the spectral norm of the Hessian of the training loss) correlates worse with generalization in these settings when using SGD, and does not correlate with it at all when using variants like Adam or Entropy-SGD.

The authors provide some preliminary explanation of the effect and propose the Bayesian prior as a more reliable generalization proxy.


**Summary Of The Review:**

# Post-rebuttal update

The authors have answered my questions and clarified that we may have some differences in interpreting their results, but I don't think it's significant enough for me to change the original score.

# Original Review

The paper presents an interesting setting in which the approximation to the Bayesian prior $P(f)$ correlates remarkably well with generalization (while flatness does not).

As I detail in my review, I find the setting to be somewhat constrained and unrealistic, and therefore the claim that the prior "is a significantly more robust predictor of generalization than flatness" is too strong, especially since there is no solid theoretical or qualitative explanation for why exactly $P(f)$ works.

Nonetheless, I still find this to be a novel and meaningful result, and look forward to future research investigating / explaining this effect further, so am inclined to accept (assuming the authors soften their claims or convince me that my concerns are overblown).

---

### Decision · Program_Chairs · 2022-01-20

**Decision:**

Reject

**Comment:**

This paper proposes the use of Bayesian prior upon initialization for predicting generalization performance of a neural network, and empirically shows that it can outperform flatness-based measures. Understanding the underlying reasons that control generalization performance on neural networks is of great theoretical and practical importance, and reviewers find efforts in this direction valuable. However, they believe the submission in current state is not ready for publication.

Specifically, ZCFx believes the setup considered in the paper does not resemble a realistic situation, which makes claims about the Bayesian prior being a more robust predictor than flatness unsubstantiated. ZCFx appreicates authors' response and clarifications, but finds the concerns unresolved. ohft believes the paper is weak in a certain aspects, such as comparing across different architectures (including number of parameters), and comparing with SAM optimizer whose goal is to find flat minima and has shown to greatly improve the generalization performance. ohft acknowledged reading authors' response but the response did not help with changing the score of the paper. r1hF has some reservations about the novelty of the work and the limited experiments, which remained unresolved. r1hF suggests that the authors revise the paper to emphasize on the author's contribution in light of the previous work.

Based on reviewers' feedback, I suggest authors to resubmit after revising the draft to address the issues raised above.